# Mapping the organisational and interventional framework for patients admitted to Hospital-at-Home for acute illness in Scandinavia – a scoping review protocol

Kristina Kock Hansen[1,2*], Maria Klitgaard Christensen[1,3], Christian Backer Mogensen[4,5], Peter Biesenbach[6], Jette Holt[7], Pia Lysdal Veje[8], Mette Elkjær[1,5], Caroline Moos[1,9]

**1** Research Center for Integrated Healthcare Region of Southern Denmark, Aabenraa, Denmark, **2** Department of Medicine, University Hospital of Southern Denmark, Lillebaelt Hospital, Vejle, Denmark, **3** Department of Clinical Research, Faculty of Health Sciences, University of Southern Denmark, Odense, Denmark, **4** Department of Emergency Medicine, University Hospital of Southern Denmark, Aabenraa, Denmark, **5** Department of Regional Health Research, University of Southern Denmark, Odense, Denmark, **6** Department of Emergency Medicine, University Hospital of Southern Denmark, Esbjerg, Denmark, **7** Infectious Disease Epidemiology & Prevention, The National Center for Infection Control (CEI), Copenhagen S, Denmark, **8** University College South Denmark, Department for Applied Health Science, Denmark, **9** Department of Clinical Research, University Hospital of Southern Denmark, Aabenraa, Denmark

* kristina.kock.hansen2@rsyd.dk

## Abstract

Hospitals in Scandinavia increasingly face an enormous pressure to manage acute emergencies in adults affected by multimorbid disabling conditions and therefore at risk of developing adverse hospitalisation outcomes such as nosocomial diseases. In this context, there is a growing political interest in the region to develop alternative models of acute emergency care such as the Hospital-at-Home (HaH), all the more so as adult patients themselves are showing a pronounced interest in HaH. We are therefore planning a scoping review, following the methodology proposed by Joanna Briggs Institute (JBI), to map the HaH components and interventions delivered to patients when managing acute emergencies in adults with that hospitalization model. We will search the databases MEDLINE, Embase and CENTRAL (Cochrane Central Register of Controlled trials) to include articles of studies on adults admitted to HaH for acute emergency care within 24 hours of getting in touch with either an adult emergency department, an out-of-hours doctor, or a general practitioner. No limitation will be placed on the search period. The electronic search will be supplemented by a grey literature search of ClinicalTrials.gov and of the archives of Ministries of Health of the Scandinavian region. The information recorded during the data extraction process will include study characteristics, participants characteristics and main review outcomes (interventions and organisational structures). Data will be synthesized narratively. This protocol has been registered with Open Science Framework (https://doi.org/10.17605/OSF.IO/K7NJS). Mapping data on HaH for the care of adult acute

**Data availability statement:** No datasets were generated or analysed during the current study. All relevant data from this study will be made available upon study completion.

**Funding:** The author(s) received no specific funding for this work.

**Competing interests:** The authors have declared that no competing interests exist.

emergencies in Scandinavia will help provide Scandinavian healthcare stakeholders with an overview of the strengths and weaknesses of different existing HaH models so that they can they can integrate that knowledge to make context-specific recommendations about and subsequently formally implement the management of adult acute emergencies in HaH in the region. Ethical approval was not required as the study does not involve human participants. Findings will be submitted for publication in a peer-reviewed Scandinavian journal and disseminated through institutional websites and LinkedIn. Results will be presented at an international conference.

## Introduction

Hospitalisation, particularly among older adults, is associated with an increased risk of nosocomial infections, physical inactivity, psychological stress and delirium [1–3]. As a response to these adverse outcomes, Hospital-at-Home (HaH) has emerged as a promising alternative. HaH has been shown to reduce the negative effects of hospitalisation, including patient distress, while also easing pressure on hospitals [4–7]. However, implementing HaH, particularly for acute illness, requires a more complex logistical and clinical setup than traditional care. In this review, we define HaH as hospital-level treatment delivered in the patient's home under the medical responsibility of a hospital team. This model is intended for managing acute illness of a severity or complexity that would otherwise require inpatient hospital admission [8].

Many existing reviews on HaH focus on specific diseases or on general admission alternatives, often overlooking the nuanced organisational structures, intervention strategies and patient characteristics involved in HaH [9–11]. In practice, HaH models vary widely in terms of patient eligibility, clinical procedures, operational logistics and healthcare professional roles. The effectiveness of HaH is also influenced by the degree of collaboration across healthcare sectors and the clarity of responsibilities among involved parties. A review from 2020 highlighted a lack of prioritisation on multidisciplinary continuity of care in HaH [12], and a systematic review underscored the need for clear guidelines regarding patient eligibility and effective cross-sector communication among staff, patients and caregivers [13].

This scoping review was initiated in response to stakeholder interest in understanding how acute HaH models are organised across Scandinavia. Despite growing international interest, the tradition of implementing HaH is less established in Scandinavian countries and evidence on HaH models within the Scandinavian context remains sparse. To date, no comprehensive reviews have addressed this region specifically. The Scandinavian healthcare system—characterised by universal, publicly funded access—differs substantially from systems in countries like the United States or Singapore, limiting the applicability of international findings.

The aim of this review is to map existing organisational structures, roles and responsibilities of healthcare personnel, and strengths and weaknesses of various HaH models specific to the Scandinavian setting. A scoping review approach was chosen due to its suitability for mapping broad areas of research, identifying

knowledge gaps and synthesising diverse study designs. We want to provide a descriptive overview of HaH organisations and interventions, including patient characteristics, and contribute with insights that can inform future practice and policy in the region.

### Review questions

1) How are HaH interventions for patients requiring emergency care organised in Scandinavia?

2) What are the characteristics of the patients offered HaH?

## Materials and methods

The proposed scoping review for evidence synthesis will be guided by methodology proposed by the Joanna Briggs Institute (JBI), including the development of this protocol to reduce the risk of impromptu decision-making [14,15]. The review will be reported according to the Preferred Reporting Items for Systematic Reviews and Meta-Analysis (PRISMA-ScR) [16]. This protocol has been registered with Open Science Framework (https://doi.org/10.17605/OSF.IO/K7NJS). The search strategy will use the Population, Concept and Context (PCC) typology [17].

The development of this review protocol was undertaken in collaboration with key stakeholders who expressed specific interest in synthesising evidence related to HaH services within the Scandinavian context. In response to the input the original search strategy was refined to ensure greater relevance to the geographical and contextual focus of the inquiry. The current search strategy reflects this adjustment, incorporating terminology and sources tailored to the Scandinavian healthcare setting.

### Inclusion and exclusion criteria

Definitions and inclusion/exclusion criteria of primary studies for this scoping review have been summarised in Table 1.

Any relevant studies identified from a foreign language will be translated using personnel from the hospital or via Chat GPT. To ensure a focus on high quality detailed evidence, we will include only original articles indexed in MEDLINE and Embase. Non primary sources such as conference abstracts, editorials, letters to the editor and preprints will be excluded as they typically lack the contextual details necessary to understand HaH organisational structures and responsibilities.

**Table 1. Definitions, inclusion and exclusion criteria based on the PCC typology.**

| | Definitions | Inclusion | Exclusion |
|---|---|---|---|
| Population | Emergency department patients Definition: Patients with the onset of an acute medical illness requiring emergency care [8] | – Adults 18 + years<br>– Initial medical contact with an emergency department (<24 hours), out-of-hours doctor or a general practitioner | – Adults requiring care described as post-surgery, long-term or end-of-life |
| Concept | HaH Definition: Treatment at home under the medical responsibility of a hospital with a service that manages acute illness at a level of severity or complexity that would require hospital admission if the intervention was unavailable [8]. | – Interventions describing admission avoidance to hospital | – Care not provided at the patient´s residence<br>– Interventions described as post-discharge, early-discharge or self-care by the patient<br>– Individuals with only a single contact with healthcare providers, for example, mobile acute care affiliated with an ED, who do not require admission. |
| Context | Scandinavia | – Studies that describe HaH in Denmark, the Faroe Islands, Finland, Greenland, Iceland, Norway, Sweden, Scandinavia, Nordic countries | – Studies from countries outside Scandinavia |

We will also exclude duplicate publications and articles that report overlapping data from the same study population without offering new findings. However, in response to stakeholders´ interests in capturing the full scope of HaH interventions in Scandinavia, we have chosen to screen some non-peer-reviewed sources for this review. We will consider experimental, quasi-experimental, observational and qualitative studies that address our review questions.

## Search strategy

Searches will be conducted using the databases MEDLINE (Ovid), Embase (Ovid) and CENTRAL (Cochrane Central Register of Controlled Trials). We will search CENTRAL rather than the full Cochrane Library as our interest is limited to peer-reviewed original studies which CENTRAL specifically indexes. The preliminary search strategy for MEDLINE is illustrated in Table 2. The search strategy will be peer-reviewed by a university librarian specialist specialising in medicine and health.

The search strategy will be fine-tuned using suggestions from the PRESS checklist over a couple of weeks with changes made collaboratively [18].

Authors will be contacted if there are any missing data from included studies. For situations whereby there are multiple publications for the same study these reports will be combined and described as the one HaH concept.

The search will be developed with three facets – acute care, Hospital-at-Home and Scandinavia. Each facet will use standardised keywords from the database's controlled vocabulary and free-text words with the Boolean operator "OR". The three facets will be combined using the Boolean operator "AND". The HaH facet was inspired by Cochrane reviews that developed search strings for this concept, albeit in two different reviews [19,20]. No restrictions will be applied to the search date to ensure the identification of all relevant literature [21]. We will use a supplementary search technique to identify any missed studies by using the Web of Science database to check references of included studies (backwards citation searching) and any relevant newer studies that have cited these included references (forward citation searching). Grey literature searches will be restricted to ClinicalTrials.gov and searches of the official Scandinavian health department websites.

## Study selection

After searching, all identified citations will be imported into Covidence [22]. Covidence is a web-based collaboration software platform that streamlines the production of systematic and other literature reviews. To support consistency in applying the inclusion criteria, a pilot process will be used that involve reviewers pausing to discuss an initial subset of records and uncertainties, and ensure screening alignment before proceeding with the full screening process. In Covidence, duplicates will be eliminated, and results will be screened independently for eligibility by two researchers in a two-step process – title/abstract screening and full-text screening. Studies not qualifying for inclusion at full-text screening will be excluded with justification. All disagreements regardless of selection stage will be resolved by consensus in accordance with the methodology proposed by JBI [15]. The search findings and the study inclusion process will be illustrated in a flow diagram following the PRISMA guidelines [16]. Grey literature from ClinicalTrials.gov will be imported into Covidence and screened with double-blinding. One author will screen any relevant website information and if considered relevant the information will be screened by a second author.

## Data charting process

The data charting process will be carried out by three researchers. One researcher (first author) will extract all data from all papers. This other two researchers will extract data blinded from the first author and any disagreements will be resolved by discussion. The extracted data will be presented in a tabular form, including study characteristics, participant characteristics and main review outcomes (see S1 Table for details of specific extraction variables). Variables to be extracted may be modified according to the findings. Although the focus of the review is organisational structures and

**Table 2. Preliminary search strategy for MEDLINE.**

| # | Query | |
|---|-------|---|
| 1 | exp Home Care Services/ | 52008 |
| 2 | exp Hospitals/ | 335091 |
| 3 | exp Hospitalization/ | 318103 |
| 4 | 2 or 3 | 614027 |
| 5 | Hospital to Home Transition/ | 71 |
| 6 | 1 and 4 | 8141 |
| 7 | 5 or 6 | 8207 |
| 8 | (hospital* adj3 home).ti,ab,kw. | 12250 |
| 9 | ((homecare or home care or home-care) and (medicalservice* or medical-service* or medical service* or health service* or health-service* or healthservice*)).ti,ab,kw. | 1286 |
| 10 | ((homebased or home-based or home based) adj (medicalcare or medical-care or medical care or medicalservice* or medical service* or medical-service* or healthservice* or health-service* or health service* or healthcare or health-care or health care or careservice* or care-service* or care service* or careprogram* or care program* or care-program*)).ti,ab,kw. | 325 |
| 11 | (homeward* or home-ward* or home ward*).ti,ab,kw. | 369 |
| 12 | ((homebased or home-based or home based) and program*).ti,ab,kw. | 7214 |
| 13 | admission avoidance.ti,ab,kw. | 187 |
| 14 | avoid* admission*.ti,ab,kw. | 296 |
| 15 | (hospital* adj2 avoid*).ti,ab,kw. | 2859 |
| 16 | (hospital* or unit* or ward* or institution*).ti,ab,kw. | 3266268 |
| 17 | 1 and 16 | 14486 |
| 18 | ((Homebased or home-based or home based) adj2 (hospitalbased or hospital-based or hospital based)).ti,ab,kw. | 95 |
| 19 | (Homehospitali?ation* or Home hospitali?ation* or Home-hospitali?ation*).ti,ab,kw. | 320 |
| 20 | Virtual ward*.ti,ab,kw. | 167 |
| 21 | Home treatment*.ti,ab,kw. | 2223 |
| 22 | ((Hospitalbased or hospital based or hospital-based) and home*).ti,ab,kw. | 2788 |
| 23 | Domiciliary hospitali?ation.ti,ab,kw. | 11 |
| 24 | Advanced care at home.ti,ab,kw. | 25 |
| 25 | (Substitute adj3 inpatient*).ti,ab,kw. | 27 |
| 26 | "hospital at home".ti,ab,kw. | 899 |
| 27 | outpatient care.ti,ab,kw. | 8340 |
| 28 | outpatient treatment*.ti,ab,kw. | 7705 |
| 29 | 7 or 8 or 9 or 10 or 11 or 12 or 13 or 14 or 15 or 17 or 18 or 19 or 20 or 21 or 22 or 23 or 24 or 25 or 26 or 27 or 28 | 55854 |
| 30 | exp Emergency Medical Services/ or exp Emergency Service, Hospital/ | 182643 |
| 31 | exp Emergency Treatment/ | 141951 |
| 32 | emergency department*.ti,ab,kw. | 147578 |
| 33 | Emergency care.ti,ab,kw. | 14177 |
| 34 | rapid response.ti,ab,kw. | 10364 |
| 35 | ((sameday or same day or same-day) and emergency care).ti,ab,kw. | 79 |
| 36 | emergency health service*.ti,ab,kw. | 452 |
| 37 | emergency treatment*.ti,ab,kw. | 5676 |
| 38 | emergency service*.ti,ab,kw. | 99978 |
| 39 | emergency patient*.ti,ab,kw. | 2573 |
| 40 | exacerbat*.ti,ab,kw. | 176032 |
| 41 | (acute* adj2 illness*).ti,ab,kw. | 13585 |
| 42 | (acute* adj2 assessment*).ti,ab,kw. | 3477 |
| 43 | (acute* adj2 service*).ti,ab,kw. | 3110 |

*(Continued)*

**Table 2.** (Continued)

| # | Query | |
|---|---|---|
| 44 | (acute* adj2 decompensat*).ti,ab,kw. | 6038 |
| 45 | acute medical.ti,ab,kw. | 4089 |
| 46 | acute disease*.ti,ab,kw. | 5048 |
| 47 | 30 or 31 or 32 or 33 or 34 or 35 or 36 or 37 or 38 or 39 or 40 or 41 or 42 or 43 or 44 or 45 or 46 | 602197 |
| 48 | exp "Scandinavian and Nordic Countries"/ | 233512 |
| 49 | Greenland/ | 2796 |
| 50 | (denmark or danish or faeroe island* or faroe island* or finland or finnish or greenland* or iceland or icelandic or icelander* or norway or norwegian* or sweden or swedish or swede* or scandinavia* or nordic countries or nordic country or Northern Europe*).ti,ab,kw. | 272329 |
| 51 | 48 or 49 or 50 | 357032 |
| 52 | 29 and 47 and 51 | 157 |

interventions, study outcomes such as mortality, readmission, length of treatment, quality of life or patient and caregiver satisfaction will also be extracted and presented in the results section.

## Data analysis

We will systematically map and narratively synthesize the data, categorize organizational descriptions, and iteratively develop organizational models based on key themes. Data will be analysed and then presented in a table format to visualize the roles, responsibilities, and collaborative relationships of actors across different sectors, as well as relevant interventions and patient characteristics.

The mapping process will follow an iterative approach, where data are continuously assessed and revised as new insights emerge from the included studies. The mapped roles and relationships will be compared to identify commonalities and variations. Based on the mapping process, a synthesis of the identified organizational models will be developed. These models will be continuously refined through an iterative process informed by new findings and theoretical considerations. The models will be described narratively and illustrated visually.

The final models will be presented with a discussion of their strengths and weaknesses. Furthermore, they will be considered in relation to practical application and their relevance for the organizational frameworks of future HaH solutions. We will not critically appraise the literature using risk of bias assessment tool, rather, all relevant studies will be included.

## Protocol deviation

Any deviations from the described protocol will be documented and transparently reported in future publications and presentations of study results. The research team will ensure that any deviations are justified, minimized, and do not compromise the integrity or objectives of the study.

## Patient and public involvement

None

## Ethical consideration

Ethical approval was not required because the study does not include participants. The study is in accordance with the Declaration of Helsinki [23] and Ethical Guidelines for Nursing Research in the Nordic Countries [24].

## Discussion

Conducting an analysis that results in models describing the different ways HaH is organised across Scandinavian countries will provide valuable insights for stakeholders involved in planning and implementing such services. By identifying and comparing key components—such as staffing structures, referral pathways, clinical responsibilities, and integration with primary and specialist care—these models can highlight effective strategies as well as context-specific challenges. This knowledge can support evidence-informed decision-making, enable adaptation to local health system structures, and ultimately facilitate more efficient and acceptable implementation of HaH tailored to the Scandinavian context.

If HaH offers the same treatment as in-hospital care, it will require significant equipment, technical devices and systems to monitor readmissions and record adverse events. It is essential to consider whether there can be similar access to services and diagnostic tools to what patients receive in the hospital [6]. Some literature describes the organisational structures of HaH programs as collaborations between hospitals and municipalities or specialised acute care teams [7,25]. Challenges arise when multiple sectors need to collaborate on patient care and share resources [26]. Additionally, patients and healthcare professionals need to feel safe and confident about the treatment received at home [12].

By limiting the literature to Scandinavian studies of admission avoidance, there is a risk of excluding important research and experiences from other countries or interventions, which may contribute to different perspectives or solutions. However, these limitations were included to ensure we assessed the degree of research and the evidence base for Scandinavian HaH interventions.

Limitations to this review include a search strategy targeting Scandinavia alone. The regional focus was considered central to the stakeholders' interests, so was deemed necessary to, e.g., avoid inflating volume. A targeted grey literature search was adopted for the same reasons. Grey literature will be sourced from ClinicalTrials.gov in relation to reporting of coming randomized control trials and websites to ensure identification of internal reports or pilot projects.

We expect to present an overview of the components of Scandinavian HaH solutions. This review will benefit other health planners and researchers considering developing HaH solutions by focusing on examples of organisational structures and components of interventions in acute HaH care.

### Dissemination plan

The review will be submitted for publication in a Scandinavian peer-reviewed journal and will be shared through institutional websites and LinkedIn. If possible, the preliminary results will be presented at an international conference.

### Supporting information

**S1 Table.  Template of data extraction sheet that will be used.**
(DOCX)

**S2 File.  PRISMA-P 2015 checklist.**
(DOCX)

### Author contributions

**Conceptualization:** Kristina Kock Hansen, Maria Klitgaard Christensen, Mette Elkjær, Caroline Moos.

**Data curation:** Kristina Kock Hansen, Caroline Moos.

**Formal analysis:** Kristina Kock Hansen, Caroline Moos.

**Investigation:** Kristina Kock Hansen, Caroline Moos.

**Methodology:** Kristina Kock Hansen, Caroline Moos.

**Project administration:** Kristina Kock Hansen.

**Software:** Caroline Moos.

**Validation:** Kristina Kock Hansen, Maria Klitgaard Christensen, Mette Elkjær, Caroline Moos.

**Writing – original draft:** Kristina Kock Hansen, Caroline Moos.

**Writing – review & editing:** Kristina Kock Hansen, Maria Klitgaard Christensen, Christian Backer Mogensen, Peter Biesenbach, Jette Holt, Pia Lysdal Veje, Mette Elkjær, Caroline Moos.

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
