## [Decision Letter · Decision Letter 0]

PONE-D-24-27648A scoping review protocol to map the interventional and organisational framework for patients admitted to hospital at home for acute illnessPLOS ONE

Dear Dr. Hansen,

Thank you for submitting your manuscript to PLOS ONE. After careful consideration, we feel that it has merit but does not fully meet PLOS ONE’s publication criteria as it currently stands. Therefore, we invite you to submit a revised version of the manuscript that addresses the points raised during the review process.

**ACADEMIC EDITOR: please revise accordingly**==============================

We look forward to receiving your revised manuscript.

Kind regards,

Zhengmao Li

Academic Editor

PLOS ONE

Reviewers' comments:

Reviewer's Responses to Questions

**Comments to the Author**

1. Does the manuscript provide a valid rationale for the proposed study, with clearly identified and justified research questions?

Reviewer #1: Partly

Reviewer #2: Partly

2. Is the protocol technically sound and planned in a manner that will lead to a meaningful outcome and allow testing the stated hypotheses?

Reviewer #1: Partly

Reviewer #2: Partly

3. Is the methodology feasible and described in sufficient detail to allow the work to be replicable?

Reviewer #1: No

Reviewer #2: No

4. Have the authors described where all data underlying the findings will be made available when the study is complete?

Reviewer #1: No

Reviewer #2: No

5. Is the manuscript presented in an intelligible fashion and written in standard English?

Reviewer #1: Yes

Reviewer #2: Yes

6. Review Comments to the Author

You may also provide optional suggestions and comments to authors that they might find helpful in planning their study.

Reviewer #1: Dear editors and authors,

Thank you for the opportunity to review the submitted manuscript outlining a protocol for a scoping review on hospital at home, and relevant interventional and organizational factors to address. The manuscript is clear and well-written, however, some revisions might enhance the clarity and readability.

Abstract

- Perhaps adding a sentence on the expected relevance and findings, to add some arguments as to why this scoping is important to carry out? Or add a sentence after the objective on what you expect to identify?

Introduction

- I suggest expanding the first argument of the introduction, adding some details on why older people are a strain to the health systems to better lay the ground for the reader.

- Some statements in the introduction is left without a reference, please add when appropriate, such as in line 79-80, line 86-87. Please check through the manuscript for other claims left without reference and add accordingly. Further, it seems unclear whether the mentioned literature is regarding all patients, adults, older adults, children, etc. Please add some details to clarify in which patients the claims are found.

- Both line 88 and line 90 starts with however, please revise.

- To clarify the role of this protocol, consider adding “proposed” in line 117 “this proposed scoping review.”

- In the aim and review question it remains unclear whether you are investigating HaH for adults, or all populations, please revise

Methods

- I suggest the design to be explicit in the opening paragraph of the methods, followed by a reference and some details on the JBI scoping in which you have chosen. This might enhance the remaining methods, making it clear to the reader what to expect.

- Please consider adding some context as to how you identified the research question. As this is a protocol, you have the space, and it is valuable for the reader to get insight into how you identified the aim and RQ

- Inclusion – please rephrase as you are not including adults, rather reports investigating adults? Please, use a concise language.

- The structure from inclusion and up until search strategy seems based on the PCC-tool for building search strategies, without you citing it. And, it seems the inclusion/exclusion only regards the patients, while the concept and context is clarify aspects of relevance to the inclusion, but it is not clear. I would suggest revising this part of the methods to explicitly state all inclusion and exclusion criteria, including those mentioned later in page 9/ line 165-169. Now, the eligibility criteria and search are described somewhat mixed.

- Is the search strategy from Embase final? If not, I would add “preliminary” or something to suggest it might change.

- Are you going to search in all databases without a time limit? Why/why not?

- Please, add a reference to Covidence, or some more data on the tool

- Line 177-178 on peer review of the search, please move to the section on the search and not on the eligibility screening

- Please cite the PRISMA ScR

- Please add some justification as to why you are extracting and planning to present data on the effects of the HaH interventions, and if relevant, please add to the aim and RQ

- Please add reference to Helsinki and the Ethical Guidelines for Nursing research in the Nordic countries.

- I would suggest adding a section on the proposed data analysis/synthesis/presentation. How are you going to summarize the findings and how are these to be presented? Perhaps removing the section named results, as you do not have any, and the current text is better placed in the methods. Then, the following section can be “Discussion” outlining the anticipated findings and their expected relevance.

Discussion

- Please, increase the amount of citations to relevant work, or rephrase so the reader can understand what are your claims and what are research supported claims

Reviewer #2: There are some elements of a scoping review protocol that are missing (e.g., data analysis, appraisal, limitations). I have the following comments.

- For clarity, I recommend formatting hospital at home as a term. Such as hospital-at-home or 'hospital at home'.

- Materials and Methods: 'Extension for Scoping Reviews' is missing from the PRISMA-ScR title.

- Inclusion criteria: There may be some typos. Perhaps the first sentence is meant to be that "In the scoping review, we will include studies that report primary research with adults..."

- Inclusion criteria: Since the Background made a compelling focus on older adults, why is this review for all adults?

- Please add a section on how data will be analysed.

- Will bias and/or quality be appraised?

- Please add a Limitations section.

7. PLOS authors have the option to publish the peer review history of their article (what does this mean? ). If published, this will include your full peer review and any attached files.

**Do you want your identity to be public for this peer review?** For information about this choice, including consent withdrawal, please see our Privacy Policy .

Reviewer #1: **Yes: ** Heidi Holmen

Reviewer #2: **Yes: ** Dr. Andrew D. Eaton

---

## [Author Response · Author response to Decision Letter 1]

4 Oct 2024

Response to Reviewers

Reviewer #1:

Thank you for the opportunity to review the submitted manuscript outlining a protocol for a scoping review on hospital at home, and relevant interventional and organizational factors to address. The manuscript is clear and well written; however, some revisions might enhance the clarity and readability.

Response: Thank you for your time and effort to improve our manuscript. We have considered your suggested revisions and would like to make the following changes after your comments. Please be aware that the line numbers we refer to in current response letter are line numbers from the unmarked version of the manuscript.

Abstract:

Perhaps adding a sentence on the expected relevance and findings, to add some arguments as to why this scoping is important to carry out? Or add a sentence after the objective on what you expect to identify?

Response:

Line 33-34: “We expect that this proposed scoping review will identify all Scandinavian Hospital-at-Home interventions that have been part of a research process.”

Introduction:

I suggest expanding the first argument of the introduction, adding some details on why older people are a strain to the health systems to better lay the ground for the reader.

Response:

We have expanded the first argument of the introduction with the following text

Line 41-46:”The Scandinavian healthcare systems are under pressure with the unprecedented numbers of people reaching retirement age and living longer with multiple chronic illnesses. These demographic changes are putting pressure on hospitals [1, 2]. Older people experience healthcare systems as fragmented with gaps in coordination. Additionally, challenges in retaining and recruiting healthcare professionals worsens the situation. Furthermore, there is a political agenda to reduce healthcare costs by reducing number of hospital beds [3] or length of stay [4].”

Some statements in the introduction is left without a reference, please add when appropriate, such as in line 79-80, line 86-87. Please check through the manuscript for other claims left without reference and add accordingly. Further, it seems unclear whether the mentioned literature is regarding all patients, adults, older adults, children, etc. Please add some details to clarify in which patients the claims are found.

Response:

Thank you for pointing that out. We have added relevant references in the text, line 57-64.

Furthermore, we have added details about which age groups of patients we are referring to:

Line 77: “Typically, patients (18+ years) receiving HaH care report high satisfaction levels [14-17].”

Line 79-80: “Both adult patients and caregivers report that HaH is beneficial as it provides dignified care while preserving the quality of life [22].”

Line 88-90: “Furthermore, adults ≥21 years old have expressed concerns about a potential decline in the quality of care and access to help if their condition worsens [22].”

Line 90-91: “This was reflected in a randomized controlled trial by Levine et al., where 157 (63%) adults declined participation in a HaH intervention [24].”

Line 95-96: “Admission avoidance appears to be an effective alternative to hospitalisation for adults with chronic obstructive pulmonary disease (COPD) exacerbations [25] or palliative care requirements [26].”

Both line 88 and line 90 starts with however, please revise.

Response:

Thank you. This is now rectified.

To clarify the role of this protocol, consider adding “proposed” in line 117 “this proposed scoping review.”

Response:

Thank you for this suggestion. We have made these changes throughout the manuscript.

In the aim and review question it remains unclear whether you are investigating HaH for adults, or all populations, please revise

Response:

Thank you for this suggestion. In line 116, we have specified much more clearly that we are interested in adult populations only.

Methods:

I suggest the design to be explicit in the opening paragraph of the methods, followed by a reference and some details on the JBI scoping in which you have chosen. This might enhance the remaining methods, making it clear to the reader what to expect.

Response:

Thank you for these clear suggestions. We have added relevant references and changed the methods section now to include following text:

Line 124-128: “The JBI Manual will guide the proposed scoping review for evidence synthesis, including developing this protocol to reduce the risk of impromptu decision-making [36, 37]. The review will be reported according to the Preferred Reporting Items for Systematic Reviews and Meta-Analysis (PRISMA-ScR) [38]. This protocol has been uploaded to Open Science Framework (DOI: 10.17605/OSF.IO/WKE7X). The search strategy will use the PCC (Population, Concept and Context) typology [39].”

Please consider adding some context as to how you identified the research question. As this is a protocol, you have the space, and it is valuable for the reader to get insight into how you identified the aim and RQ

Response:

Thank you for pointing out this out. We have included the following text in the introduction:

Line 64-75: “Evidence on HaH programs within the Scandinavian context is limited, as Scandinavia has a less developed tradition of implementing HaH services. Additionally, Scandinavian healthcare system is characterised by free, publicly funded hospitals available to all citizens through taxation, making it challenging to draw direct comparisons with HaH solutions in countries like USA or the UK.

Our study's research question was driven by Danish healthcare administrators request for evidence on HaH solutions that ensure safety and effectiveness for patients and healthcare professionals. While Scandinavia has recently implemented some initiatives focusing on home-based telehealth and HaH services, there is limited published research on these efforts. Moreover, the literature appears to lack a comprehensive overview of various HaH solutions, including key aspects such as organisational structures, patient demographics and designated responsibility of care [3].”

Inclusion – please rephrase as you are not including adults, rather reports investigating adults? Please, use a concise language.

Response:

Thank you. We have removed the text and made a table instead with inclusion and exclusion criteria (also based on your next comment), page 8-9.

The structure from inclusion and up until search strategy seems based on the PCC-tool for building search strategies, without you citing it. And, it seems the inclusion/exclusion only regards the patients, while the concept and context is clarify aspects of relevance to the inclusion, but it is not clear. I would suggest revising this part of the methods to explicitly state all inclusion and exclusion criteria, including those mentioned later in page 9/ line 165-169. Now, the eligibility criteria and search are described somewhat mixed.

Response:

Thank you for your input. We decided that it would be clearer, if the inclusion and exclusion criteria for each block were presented in table format. Please see Table 2, page 9. We have also included the relevant reference for the PCC tool (line 128). We hope the descriptions of the search and subsequent eligibility criteria are now clearer.

Is the search strategy from Embase final? If not, I would add “preliminary” or something to suggest it might change.

Response:

Thank you for this suggestion. Yes, it was preliminary and is added in line 132 and in the heading of Table 1. We have decided to add an extra block to ensure that we only include Scandinavian research. This search strategy is therefore being amended (see Table 1, page 6-7).

Are you going to search in all databases without a time limit? Why/why not?

Response:

We have decided not to apply a time limit to the database search. Therefore, all databases will be search without a time limit. The reason behind this is to ensure that we do not increase the likelihood of bias, by excluding relevant evidence. (reference: Higgins JPT, Thomas J, Chandler J, Cumpston M, Li T, Page MJ, Welch VA (editors). Cochrane Handbook for Systematic Reviews of Interventions version 6.5 (updated August 2024). Cochrane, 2024.)

We have included the following information:

Line 140-141: “No restrictions will be applied to the search date to ensure the identification of all relevant literature [40].”

Please, add a reference to Covidence, or some more data on the tool

Response:

Thank you for identifying this. We have added a reference (line 155).

Line 177-178 on peer review of the search, please move to the section on the search and not on the eligibility screening

Response:

Thank you for identifying this. We have now moved the sentence to the appropriate subheading (line 141).

Please cite the PRISMA ScR

Response:

Thank you. This reference is now added in line 126.

Please add some justification as to why you are extracting and planning to present data on the effects of the HaH interventions, and if relevant, please add to the aim and RQ

Response:

Thank you for pointing this out. We would like to get a picture of the research based behind Hospital-at-Home interventions in Scandinavia. It is useful when planning a similar intervention to look at the effect outcomes measured. Therefore, the following text has now been incorporated into the last sentence of the introduction

Line 114-117: “Alternative solutions are imperative in anticipation of the future strain on the healthcare system. This proposed scoping review aims to map Scandinavian organisational structures, interventions and describe population characteristics as well as effect outcomes of adult patients receiving HaH care for an acute medical illness that would otherwise require acute hospital inpatient care for a limited period.”

Please add reference to Helsinki and the Ethical Guidelines for Nursing research in the Nordic countries.

Response:

Thank you. These references are now added (line 183-184).

I would suggest adding a section on the proposed data analysis/synthesis/presentation. How are you going to summarize the findings and how are these to be presented? Perhaps removing the section named results, as you do not have any, and the current text is better placed in the methods. Then, the following section can be “Discussion” outlining the anticipated findings and their expected relevance.

Response:

Thank you for this. We have now removed the result section and have added a “Proposed data presentation” section with a table indicating how we want to present our results. See line 174-179, page 10.

Discussion:

Please, increase the amount of citations to relevant work, or rephrase so the reader can understand what are your claims and what are research supported claims

Response:

Thank you for the input. The discussion section has been updated with citations to support research-based claims. You can see these changes in the discussion section between line 187 and line 200.

Reviewer #2:

There are some elements of a scoping review protocol that are missing (e.g., data analysis, appraisal, limitations). I have the following comments.

For clarity, I recommend formatting hospital at home as a term. Such as hospital-at-home or 'hospital at home'.

Response:

Thank you. We have formatted again using Hospital-at-Home as the term.

Materials and Methods: 'Extension for Scoping Reviews' is missing from the PRISMA-ScR title.

Response:

Thank you. This has now been amended in line 125-126.

Inclusion criteria: There may be some typos. Perhaps the first sentence is meant to be that "In the scoping review, we will include studies that report primary research with adults..."

Response:

We have adjusted the inclusion and exclusion criteria after both reviewers´ comments and identified the typos and corrected them. We have had a native English speaker editing the protocol. See Table 2, page 8 and 9.

Inclusion criteria: Since the Background made a compelling focus on older adults, why is this review for all adults?

Response:

Very relevant question, thanks. We are interested in identifying any Scandinavian HaH and later assessing if they are relevant for older adults.

Please add a section on how data will be analysed.

Response:

Please see Line 178-179 (Table 3). We expect the data to be very heterogeneous so a narrative summary will be the most relevant.

(Reference: Popay, J., H. Roberts, A. Sowden, et al. 2006. “Guidance on the Conduct of Narrative Synthesis in Systematic Reviews.” A Product From the ESRC Methods Programme Version 1, no. 1: b92.https://citeseerx.ist.psu.edu/document?repid=rep1&type=pdf&doi=ed8b23836338f6fdea0cc55e161b0fc5805f9e27).

Will bias and/or quality be appraised?

Response:

As this is a scoping review, bias and quality will not be appraised, but may be relevant for any later systematic review.

(Reference: Munn Z et al. Systematic review or scoping review? Guidance for authors when choosing between a systematic or scoping review approach. BMC Med Res Methodol. 2018 Nov 19;18(1):143. DOI: 10.1186/s12874-018-0611-x).

Please add a Limitations section.

Response:

We agree that a limitation section would be relevant. We have added the following text to the discussion section:

Line 202-208: “By limiting the literature to Scandinavian studies of admission avoidance, there is a risk of excluding important research and experiences from other countries or interventions, which may contribute to different perspectives or solutions. However, these limits ensure that HaH solutions are more homogeneous. The narrow grey literature search may have excluded relevant internal reports, white papers, conference papers, local initiatives or innovative pilot projects. However, these limitations were included to ensure we assessed the degree of research and the evidence base for Scandinavian HaH interventions.”

---

## [Decision Letter · Decision Letter 1]

PONE-D-24-27648R1

Mapping the organisational and interventional framework for patients admitted to Hospital-at-Home for acute illness – a scoping review protocol

PLOS ONE

Dear Dr. Hansen,

Thank you for submitting your manuscript to PLOS ONE. After careful consideration, we have decided that your manuscript does not meet our criteria for publication and must therefore be rejected.

Specifically:

**see comments**

I am sorry that we cannot be more positive on this occasion, but hope that you appreciate the reasons for this decision.

Kind regards,

Zhengmao Li

Academic Editor

PLOS ONE

Additional Editor Comments (if provided):

the reviewer is angry with the revision and he insist to reject.sorry

Reviewers' comments:

Reviewer's Responses to Questions

**Comments to the Author**

1. Does the manuscript provide a valid rationale for the proposed study, with clearly identified and justified research questions?

Reviewer #1: Partly

Reviewer #2: No

2. Is the protocol technically sound and planned in a manner that will lead to a meaningful outcome and allow testing the stated hypotheses?

Reviewer #1: Yes

Reviewer #2: No

3. Is the methodology feasible and described in sufficient detail to allow the work to be replicable?

Reviewer #1: Yes

Reviewer #2: No

4. Have the authors described where all data underlying the findings will be made available when the study is complete?

Reviewer #1: Yes

Reviewer #2: No

5. Is the manuscript presented in an intelligible fashion and written in standard English?

Reviewer #1: Yes

Reviewer #2: No

6. Review Comments to the Author

You may also provide optional suggestions and comments to authors that they might find helpful in planning their study.

Reviewer #1: Dear editor and authors,

Thank you for the opportunity to review the revised version of this scoping review protocol on Hospital-at-Home (HaH). The authors have clarified their manuscript in a thorough manner with well-founded revisions based on the comments from the reviewer.

One aspect has, however, been added to the revised version without being specified in the initial version of the manuscript, neither asked for by the reviewers. That is the narrow focus on the Scandinavian perspective, absent in the initial version. I see how the introduction has been revised according to the Scandinavian focus, but I was surprised of this specification of inclusion. I also see how this narrow focus might conflict with the broad focus on the time perspective of the reported research. Would the authors share their rationale for the Scandinavian specific focus?

I acknowledge the risk of selection bias if adding a time limit; however, the context of specialized hospital-based services has changed greatly the last some-thirty years, in addition to the specific application of HaH in these settings – with a supplement of digital homebased services or communication. I cannot see the consistence in the authors argumentation between a broad time and a narrow context, and I worry the scoping will be in-consistent and result in a less stringent scoping overview.

I would be interested in the reflections of the authors in this matter.

I have no further comments, and I acknowledge the remaining revisions made from the authors.

Reviewer #2: The authors have provided an insufficient revision that raises concerns. A Scandinavian focus was not clear from the initial version, raising questions of the study’s rigour. It is entirely inappropriate to not present an analytic plan in a protocol. A brief mention of a “narrative summary” is inappropriate. If the authors will not assess bias and/or quality, they cannot adequately present a synthesis of the literature.

7. PLOS authors have the option to publish the peer review history of their article (what does this mean? ). If published, this will include your full peer review and any attached files.

**Do you want your identity to be public for this peer review?** For information about this choice, including consent withdrawal, please see our Privacy Policy .

Reviewer #1: **Yes: ** Heidi Holmen

Reviewer #2: No

- - - - -

---

## [Author Response · Author response to Decision Letter 2]

7 Dec 2024

Response to Reviewers

We would be grateful if you could please revise the manuscript to respond to the comments raised by the Academic Editor and reviewers, and resubmit your manuscript files online once the necessary revisions are complete. Please accompany this resubmission by a detailed rebuttal letter that includes a point-by-point response to the concerns raised by the reviewers and Academic Editor and details on the revisions carried out on the manuscript since its original submission. Please also submit a “Track Changes” version of the MS file in the File Inventory. You may submit some of the files provided in your appeal request if appropriate.

We suggest that you take particular care to justify the changes to the focus of your review since your last submission. One of the goals of Study Protocols is to increase transparency. Therefore, we would recommend an explicit statement within the manuscript detailing the reasons for changes to the methods whilst this protocol underwent review, and justifications for doing so.

Response: Thank you for considering our appeal request and the opportunity for our protocol to be reconsidered. We appreciate the opportunity to provide clarity to the issues raised. We have revised the manuscript according to the comments raised by the Academic Editor and reviewers and included a point-by-point response to the concerns raised. As suggested, we have focused specifically on a transparent explanation of the change in search strategy that narrowed references to Scandinavia. The comments from reviewers have also resulted in us specifically addressing the chosen “analytical” approach narrative synthesis.

Reviewer #1:

Dear editor and authors,

Thank you for the opportunity to review the revised version of this scoping review protocol on Hospital-at-Home (HaH). The authors have clarified their manuscript in a thorough manner with well-founded revisions based on the comments from the reviewer. One aspect has, however, been added to the revised version without being specified in the initial version of the manuscript, neither asked for by the reviewers. That is the narrow focus on the Scandinavian perspective, absent in the initial version. I see how the introduction has been revised according to the Scandinavian focus, but I was surprised of this specification of inclusion. I also see how this narrow focus might conflict with the broad focus on the time perspective of the reported research. Would the authors share their rationale for the Scandinavian specific focus?

Response: Yes, the Scandinavian perspective was added to the revised version of the protocol. Our stakeholders were only interested in a review of HaH within the Scandinavian region. Our research team discussed that studies identified from other healthcare systems were perhaps not useful as they could not be easily implemented into a Danish context. Furthermore, as the protocol was not yet published, we were of the opinion that we had flexibility to make refinements to ensure the review remained as comprehensive and relevant as possible. Therefore, this adjustment aimed to ensure that our findings could directly inform the development and implementation of acute HaH programs in a context relevant to Denmark. Denmark´s healthcare is characterized by universal access, strong integration between primary and secondary care similar to other Scandinavian countries.

We have carefully documented and justified all modifications to maintain transparency and integrity. We hope this explanation clarifies our rationale and demonstrates our commitment to maintaining methodological rigor.

Reviewer 1: I acknowledge the risk of selection bias if adding a time limit; however, the context of specialized hospital-based services has changed greatly the last some-thirty years, in addition to the specific application of HaH in these settings – with a supplement of digital homebased services or communication. I cannot see the consistence in the authors argumentation between a broad time and a narrow context, and I worry the scoping will be in-consistent and result in a less stringent scoping overview. I would be interested in the reflections of the authors in this matter.

We acknowledge that hospital-based services and the application of HaH programs have evolved significantly over the past three decades. However, we are still interested in capturing all relevant developments within Scandinavia. HaH is a relatively new area of focus in Scandinavia and we were interested in all studies regardless of their age. Narrowing the timeframe could risk excluding research describing the initial implementation of HaH and the thoughts behind it. We hope this explanation clarifies our rationale and demonstrates our commitment to producing a focused, comprehensive, and impactful review.

Reviewer #2:

The authors have provided an insufficient revision that raises concerns. A Scandinavian focus was not clear from the initial version, raising questions of the study’s rigour.

Response: See response above to reviewer 1

Reviewer 2: It is entirely inappropriate to not present an analytic plan in a protocol. A brief mention of a “narrative summary” is inappropriate. If the authors will not assess bias and/or quality, they cannot adequately present a synthesis of the literature.

Response:

Thank you for raising this important point and offering us the opportunity to explain in more depth. We will elaborate on our rationale for not including bias and quality assessments in this letter and will in our manuscript incorporate a more in-depth explanation of narrative synthesis.

As the primary aim of this scoping review is to map the breadth and characteristics of HaH solutions, including patient populations, interventions and organizational structures, it aligns with established guidance that scoping reviews are not intended to synthesize or critically appraise evidence as a systematic review would. Instead, they aim to provide a descriptive overview to identify knowledge gaps and understand the extent of the available literature (Arksey & O’Malley, 2005).

The inclusion of bias and quality assessments is not a standard requirement for scoping reviews, as noted in the JBI guidance on scoping reviews (Aromataris et al. 2024). Scoping reviews often focus on descriptive mapping rather than synthesizing results or evaluating the certainty of evidence. For example, it is appropriate to extract and descriptively map results without attempting to assess the certainty or quality of individual studies (Munn et al., 2018). This approach is particularly relevant for our review, which seeks to understand variations in HaH models and the contextual factors, rather than to draw definitive conclusions about intervention effectiveness.

For this review, we have opted for a narrative synthesis approach to systematically map the data while maintaining transparency and clarity in our methods. Please see line 181-185 in the manuscript. To support our methodological choice, we refer to Popay et al.'s (2006) guidance on narrative synthesis in systematic reviews, which highlights that narrative synthesis is a robust and widely accepted method, particularly when handling diverse and heterogeneous evidence bases. The approach allows for a structured and transparent synthesis of findings without requiring quality assessments to be conducted as part of the process. Moreover, our narrative synthesis will be conducted systematically, emphasizing consistent reporting of patient characteristics, intervention types and organizational models.

We hope this response adequately addresses your concerns and provides clarity on why our approach, without bias or quality assessments, is methodologically sound and consistent with best practices for scoping reviews. Thank you again for your insightful comments, which have allowed us to strengthen our protocol.

References

Arksey, H., & O’Malley, L. (2005). Scoping studies: towards a methodological framework. International Journal of Social Research Methodology, 8(1), 19–32. https://doi.org/10.1080/1364557032000119616

Aromataris E, Lockwood C, Porritt K, Pilla B, Jordan Z, editors. JBI Manual for Evidence Synthesis. JBI; 2024. Available from: https://synthesismanual.jbi.global. https://doi.org/10.46658/JBIMES-24-01

Munn, Z., Peters, M.D.J., Stern, C. et al. Systematic review or scoping review? Guidance for authors when choosing between a systematic or scoping review approach. BMC Med Res Methodol 18, 143 (2018). https://doi.org/10.1186/s12874-018-0611-x

Popay, Jennie et al. “Guidance on the conduct of narrative synthesis in systematic Reviews. A Product from the ESRC Methods Programme. Version 1.” (2006).

---

## [Decision Letter · Decision Letter 2]

PONE-D-24-27648R2Mapping the organisational and interventional framework for patients admitted to Hospital-at-Home for acute illness – a scoping review protocolPLOS ONE

Dear Dr. Hansen,

Thank you for submitting your manuscript to PLOS ONE. After careful consideration, we feel that it has merit but does not fully meet PLOS ONE’s publication criteria as it currently stands. Therefore, we invite you to submit a revised version of the manuscript that addresses the points raised during the review process.

We look forward to receiving your revised manuscript.

Kind regards,

Mickael Essouma, M. D.

Academic Editor

PLOS ONE

Journal Requirements:

Comments from PLOS Journal Office: Please note that for scoping review protocols, PLOS One requires a copy of the PRISMA-P checklist that you have provided and not the PRISMA-ScR checklist that is being requested by Reviewer 6. You can state that this is due to a journal requirement in your response to the reviewers. 

Additional Editor Comments (if provided):

I have made a few comments in the attached pdf.

Reviewers' comments:

Reviewer's Responses to Questions

**Comments to the Author**

1. Does the manuscript provide a valid rationale for the proposed study, with clearly identified and justified research questions?

Reviewer #2: No

Reviewer #3: No

Reviewer #4: Yes

Reviewer #5: Yes

Reviewer #6: Partly

Reviewer #7: Yes

Reviewer #8: Yes

2. Is the protocol technically sound and planned in a manner that will lead to a meaningful outcome and allow testing the stated hypotheses?

Reviewer #2: No

Reviewer #3: Partly

Reviewer #4: Partly

Reviewer #5: Yes

Reviewer #6: Partly

Reviewer #7: Yes

Reviewer #8: Partly

3. Is the methodology feasible and described in sufficient detail to allow the work to be replicable?

Reviewer #2: No

Reviewer #3: No

Reviewer #4: Yes

Reviewer #5: Yes

Reviewer #6: No

Reviewer #7: Yes

Reviewer #8: Yes

4. Have the authors described where all data underlying the findings will be made available when the study is complete?

Reviewer #2: No

Reviewer #3: No

Reviewer #4: Yes

Reviewer #5: Yes

Reviewer #6: Yes

Reviewer #7: Yes

Reviewer #8: Yes

5. Is the manuscript presented in an intelligible fashion and written in standard English?

Reviewer #2: Yes

Reviewer #3: No

Reviewer #4: Yes

Reviewer #5: Yes

Reviewer #6: Yes

Reviewer #7: Yes

Reviewer #8: Yes

6. Review Comments to the Author

You may also provide optional suggestions and comments to authors that they might find helpful in planning their study.

Reviewer #2: I recommended rejection of a prior version of this manuscript because there was no analytic plan presented. Analytic plans are a requirement for scoping review protocols. There is still no analytic plan in the manuscript.

Reviewer #3: RE Manuscript Number PONE-D-24-27648R2: The authors have presented a protocol for a scoping review on Hospital-at-Home structures and interventions for adults. Please find my detailed comments below:

1) I was unable to review the protocol on OSF. The authors should note that the OSF registration needs to outline the key information of the protocol clearly and transparently. Simply registering the title may not provide enough context for other researchers. You may refer to this for further information: https://osf.io/ym65x.

2) The abstract's transition from a general objective to focus on Scandinavian Hospital-at-Home is abrupt. The authors need to clearly state their focus in their objective.

3) The methods of the abstract need to outline the date range of the study. Furthermore, focusing only on peer-reviewed articles is not in line with the methodology of a scoping review.

4) The consistency and cohesion between the topics presented in the introduction needs to be revised. For example, after the paragraph discussing the origin of the research question, readers might expect a more direct continuation of that theme. However, the subsequent paragraphs introduce a new concept, which could disrupt the flow of the introduction.

5) The review question needs to implement the PCC framework. At the time of writing this peer review, the review question is rather broad and difficult to navigate and understand. The authors may consider reformulating the research question in three distinct segments.

6) I concur with the reviewers and editor that the focus on Scandinavian region may be too narrow for the purpose of a scoping review. However, since the authors are interested only in the Scandinavian region, I would recommend that they target the grey literature that are more relevant to that specific region. This source may provide useful databases that the author may consider to search and state in their protocol: https://www.hvl.no/en/library/find-databases-and-other-resources/.

7) I would recommend the authors to summarize their data extraction in a table and present them to the readers. This will also save their time when extracting the results in the review process once you have preplanned data extraction table. The authors can then use the freed space in their protocol to articulate on how they will organize the data extraction. They can elaborate the process on data extraction, namely the minimum number of people involved, and how disagreements will be resolved.

8) Instead of the title 'Proposed data synthesis' the authors may consider 'Data analysis'.

9) I concur with the authors regarding risk of bias assessment being inappropriate in the context of a scoping review. Still, I would encourage the authors to include a statement in their protocol that they will not critically appraise the literature using any risk of bias assessment tools.

10) Under the Discussion section, the authors have stated that by limiting the results to Scandinavian studies, the output will be homogenous. I would argue that some readers may misinterpret this from a statistical perspective, as in the homogeneity of data will allow the authors to conduct certain statistical analysis. I would recommend the authors using other terminologies to discuss this shortcoming and link this to what was expected from the Danish healthcare administrators.

11) I would suggest the authors clearly state their dissemination plans once the review is over. This could include presentation of the findings in a peer-reviewed journal.

12) PRISMA-P was not designed to report scoping review protocols. I would recommend the authors delete that in case of future submissions.

Reviewer #4: Dear Respectable Authors

Thank you for considering a great area of research related to acute illness. Your responses to previous reviewer is well but your manuscript needs some revisions as follows;

- Abstract, methods section, please add some information regarding charting methods.

- Based on the PRISMA-ScR, the eligibility criteria is the first step in reporting a scoping review. Please set this subheading as the first item in the methods section, after protocol and registration and then mention the search.

- In scoping review we use the "data charting process" instead on "data extraction", then, it is better to use a separate subheading including "data charting process and data items" (Items 10 and 11 of PRISMA ScR).

Cheers

Reviewer #5: This is a useful topic and appropriately justified. I understand the limit to the Scandinavian perspective if the research is being undertaken to satisfy particular stakeholders. I believe the methods that are selected are appropriate and have been adequately justified, particularly with regard to the time span and setting.

That said, more detail and transparency is needed for the literature search. It is not clear how exactly the peer review will be undertaken (PRESS peer review https://pubmed.ncbi.nlm.nih.gov/27005575/ is the most commonly used method) and how the authors commit to addressing any concerns. Some errors or limitations of the current strategy are noted - line 48 should combine lines 29 and 47, not 29 and 46. There are changes in language and truncation that would also enhance the strategy. For example, the geographic filter doesn’t incorporate regional areas, capital cities, citizens (e.g., Danish, Norwegian) and there are alternate spellings for some locations such as the Faeroe Islands. It would also be helpful to see the number of hits per line. There is a small error in reference to the grey literature source, which is ClinicalTrials.gov (i.e., plural). MEDLINE should be written in block caps.

The authors state that deduping will be done within Covidence. If work is not yet complete, I would recommend using a reference manager software such as EndNote for initial deduping, with final/refined deduping done in Covidence. It is challenging to confirm duplicates within Covidence compared to the flexibility available in most reference manager software.

I am not clear who is doing the data extraction. This should be stated in the methods.

There is an inaccuracy in the example regarding HaH solutions, where the authors suggest that it is challenging to compare the Scandinavian system with that of the UK. I believe the UK healthcare system and its hospitals are publicly funded, so another example/country alongside the US should be highlighted.

The manuscript flows and is easy to understand although some sentences are a bit stilted, no doubt because the authors are writing in their non-native tongue (for which they are to be congratulated). E.g., Line 51 – “political agency to reduce healthcare costs by reducing number of hospital beds…” requires “the” before number; Line 71 – “Additionally, Scandinavian healthcare system is characterised…” requires “the” before Scandinavian. Line 118 is also a bit unclear – “to design effective, patient and work safe acute HaH programs” – suggest instead wording as “to design effective HaH programs that are safe for both patients and providers (if that is the intended meaning).

I would suggest some small copy edits in the manuscript (e.g., mathematical symbols should be written as text in the manuscript, so change “<” to less than, “18+” to 18 years of age and older (or something similar, e.g., patients over the age of 18). “Adults >= 21 years old” should be written as “Adults aged 21 or older”. In the abstract, “out of hours” should be hyphenated for “out-of-hours doctor or general practitioner”. There is a possessive missing at the end of administrators’ in line 76 “driven by Danish healthcare administrators request”. I would substitute “although” for “but” in the line “it can be physically and mentally demanding… Free text should be hyphenated in the line “free-text words with the Boolean operator…”.

Reviewer #6: Dear Authors and Editor,

Thank you for considering me as a reviewer for this manuscript titled “Mapping the organisational and interventional framework for patients admitted to Hospital-at-Home for acute illness – a scoping review protocol.” Below are my observations:

General Comments

I understand that this is the third (?) round of comments on this manuscript, which is a scoping review protocol. I have read the previous reviewers’ comments as well as the authors’ responses. I understand that the initial version of the manuscript did not specify a contextual framework (Scandinavian), but the authors later modified this aspect in the subsequent versions. On this point, I agree with the authors regarding the need to reassess the protocol and adapt it to the research needs. However, it is important to understand the editorial process, as reviewers evaluate an initial version and provide comments based on that version. When modifications are made that were not requested, this can create confusion, as it complicates the understanding of what the authors actually intend to investigate.

That said, regarding the restriction to the Scandinavian context, the authors should consider the following:

1. In the introduction, the authors mention that research on this topic in the Scandinavian context is limited. Would this not contradict the decision to limit the review exclusively to the Scandinavian context? Additionally,

2. Why is the Danish healthcare system so different from other systems that it could not benefit from research produced in other contexts? If research in the Scandinavian context is limited, knowing what is happening beyond this context could provide the authors with a more comprehensive overview of the strategies being used elsewhere and how they could be implemented in their setting. The authors could consider formulating a sub-question to address the Scandinavian context specifically, allowing the main research question to remain open and better aligned with the scoping review methodology.

• Introduction

The introduction is appropriate; however, there are pieces of information that appear disconnected from the text or are not discussed in context. For example:

Lines 52-53: “Hospitalization is associated with a higher risk of nosocomial infections, inactivity, stress and delirium, particularly in older people [5-7].”

Additionally, in the introduction, the authors frequently refer to the older adult population; however, the study will include individuals aged 18 and over. I suggest clarifying this point.

• The authors mention (lines 51-52) that there are policies aimed at reducing the number of hospital beds and the length of stay. In this regard, the potential of HaH as an alternative is not adequately discussed, especially considering that a more complex logistical structure is required to provide hospital-level care at home, particularly in cases of acute illness (as proposed in this review).

• Another point that caught my attention, which I suggest reviewing, is citation [29] referring to “older patients with acute cardiac failure.” While the information is accurate, upon reviewing the cited article, it appears to be a retrospective study. Given the methodological limitations of retrospective studies, it may not be sufficient to generalize the findings to the broader population. Care should be taken when using individual studies to make generalizations, especially when there are limitations in their design. For instance, if we consider an older adult with decompensated heart failure, we should question whether, in an acute situation, they would receive better care at home than in a hospital. This is particularly relevant in healthcare systems with limited financial and human resources. Therefore, while the information is not incorrect, it requires further discussion.

• Methods

The authors state that the protocol development adhered to the PRISMA-ScR guidelines (correct); however, the attached document refers to PRISMA-P, which is confusing. Is there an explanation for this? If not, PRISMA-ScR should replace PRISMA-P.

• The PCC acronym is central to scoping reviews; therefore, its structure should be consistently reflected throughout the methodology section. How is PCC integrated into the search strategy?

• Additionally, why are only three databases included? While Cochrane recommends these three as essential, in scoping reviews, the goal is to map the phenomenon extensively. Therefore, it is advisable to include additional databases (e.g., Scopus, Web of Science, CINAHL). I suggest reconsidering this. Similarly, regarding grey literature—why is only one database (clinical trials) included?

• Please add a mention of the controlled vocabularies that will be used.

• For articles that are not fully accessible, what is the plan to obtain them?

• Considering the limited existing evidence, do the authors believe that this will be adequately reported in the studies? If not, how will they evaluate this aspect: "Initial medical contact with an emergency department (<24 hours), out-of-hours doctor, or a general practitioner" (line 158)?

• In the study selection process, I suggest adding information on the agreement between reviewers before starting the screening process. Please refer to the relevant section in the JBI Manual.

• Lines 166-167: “The search findings and the study inclusion process will be illustrated in a flow diagram following the PRISMA extension for Scoping Review guidelines.” This is incorrect. There is no extension of the PRISMA flow diagram for scoping reviews. The flow diagram is the same for all types of reviews; what changes is the checklist. Please revise this.

• How will the grey literature be reviewed?

• How are the extraction variables related to the review topic? How were they chosen? Did the authors adapt the extraction form from other studies, or was it created specifically for this review? Please consult the JBI Manual.

• The authors describe how the data will be presented but do not specify how it will be analyzed. Will content analysis be used? Will there be a specific analytical framework? Please consult the JBI Manual.

Final Comment

The manuscript is interesting and has the potential to contribute to the scientific community. However, to ensure that the proposed review is framed within appropriate theoretical and methodological foundations, I suggest carefully reading the manuscript alongside the comments provided. Remember that review studies should offer sufficient detail to allow readers to understand how the review process was conducted, even more so when it is a protocol article. I suggest that the authors carefully review their work and provide greater detail on each of the procedures to be carried out. Particular attention should be paid to the JBI guideline.

Reviewer #7: Review of the Manuscript: “Mapping the organisational and interventional framework for patients admitted to Hospital-at-Home for acute illness – a scoping review protocol”

Thank you for the opportunity to review this manuscript. I appreciate the effort the authors have put into developing this scoping review protocol. Below are my suggestions for improvement:

1. Title, Abstract, and Research Question: I recommend explicitly including the Scandinavian context in the title, abstract, and research question to ensure transparency regarding the focus of the scoping review (ScR) and its PCC framework.

2. Grey Literature in the Search Strategy: On page 7, the search strategy section discusses the inclusion of grey literature. However, this is not mentioned in the abstract. I suggest incorporating this aspect in lines 147 and 148 to maintain consistency.

3. Justification for ScR Methodology: A more robust justification for selecting the ScR methodology would strengthen the Materials and Methods section. This justification could also serve as a foundation for explaining why quality assessment is not conducted for the included studies. I suggest adding this explanation at the end of the search strategy section, as readers unfamiliar with the ScR methodology might question the absence of a quality assessment.

4. Discussion on Grey Literature Inclusion: The rationale for the limited inclusion of grey literature is not entirely clear in the discussion section. If a justification is necessary, I recommend clarifying this in line 216, particularly regarding the phrase "to assess the degree of research," which could be more precise. Additionally, if the objective is to present an overview of Scandinavian Hospital-at-Home (HaH) components (as stated in line 218), would it not be relevant to include what was excluded in lines 214 and 215? A clearer explanation of these decisions would enhance the discussion.

Reviewer #8: The research topic is of practical significance and has potential value for improving the HaH service system and guiding clinical practice. However, there are also some deficiencies, which affect the depth, breadth, and rigor of the research and need to be further improved.

1. Duplication in background elaboration: In the introduction part, background information such as the pressure on the healthcare system and aging is repeatedly emphasized, resulting in redundant content. It can be streamlined and integrated to highlight the key points and avoid repetition.

2. Limited scope: The research only focuses on the Scandinavian region and excludes studies from other countries, which may lead to the omission of valuable experiences and viewpoints. Is it possible to expand the search scope, include internationally representative studies, compare the differences in the HaH models in different regions, and provide a more comprehensive reference for the Scandinavian region?

3. It is recommended to conduct stratified comparison: Considering that factors such as the severity of patients' conditions and underlying diseases may affect the research results, further stratified comparison can be carried out. For example, control groups can be set up by stratifying according to disease types (such as cardiovascular diseases, respiratory diseases, etc.) and the severity of the disease (mild, moderate, severe), ensuring that the research results are more targeted and reliable, and can accurately reflect the effectiveness of the HaH model under different circumstances.

4. Insufficient categorization of intervention measures: The description of HaH intervention measures is broad and lacks detailed classification. It should be further refined, for example, classified according to disease types, intervention intensity, service content, etc., to facilitate in - depth analysis of the effectiveness of different intervention measures and their applicable populations.

7. PLOS authors have the option to publish the peer review history of their article (what does this mean? ). If published, this will include your full peer review and any attached files.

**Do you want your identity to be public for this peer review?** For information about this choice, including consent withdrawal, please see our Privacy Policy .

Reviewer #2: No

Reviewer #3: **Yes: ** Amin Sharifan

Reviewer #4: **Yes: ** Morteza Arab-Zozani

Reviewer #5: No

Reviewer #6: No

Reviewer #7: No

Reviewer #8: No

---

## [Author Response · Author response to Decision Letter 3]

10 Apr 2025

Response to reviewers

Dear reviewers #2, #3, #4, #5,#6, #7, #8 and Editorial team,

Thank you for your valuable feedback on our scoping review protocol article. We truly appreciate the effort invested in providing these insights, which will strengthen the final review. In this response letter and revised manuscript, we have outlined how we have addressed your concerns and the points that needed extra clarification. While the revisions may seem complex (due in part to the involvement of multiple reviewers), we have tried to structure our responses to ensure each reviewer has their concerns explicitly addressed, with clear indications of the changes made to the manuscript. We have also highlighted criticisms that multiple reviewers brought to our attention. We have included the page and line numbers where the adjustments can be made but decided that this was most simple for the reviewers if it referred to the “clean” manuscript. We hope this simplifies the revision process.

2 Reviewer #2

2.1 I recommended rejection of a prior version of this manuscript because there was no analytic plan presented. Analytic plans are a requirement for scoping review protocols. There is still no analytic plan in the manuscript.

Response: We appreciate your commitment to ensuring the methodological rigor of this scoping review protocol. We acknowledge your concern regarding the absence of an explicit analytic plan in our previous submission. Based on your feedback, we have included the following additions in the manuscript to outline more specifically our analytic approach (line 157-172, page 9). This comment is similar to Reviewer #6 (see 6.17)

“We will systematically map and synthesize the data, categorize organizational descriptions, and iteratively develop organizational models based on key themes. All included studies and documents will be systematically reviewed to identify descriptions of organizational structures, interventions, and patient characteristics related to the research question. Data will be extracted and then presented in a table format to visualize the roles, responsibilities, and collaborative relationships of actors across different sectors, as well as relevant interventions and patient characteristics.

The mapping process will follow an iterative approach, where data are continuously assessed and revised as new insights emerge from the included studies. The mapped roles and relationships will be compared to identify commonalities and variations. Based on the mapping process, a synthesis of the identified organizational models will be developed. These models will be continuously refined through an iterative process informed by new findings and theoretical considerations. The models will be described narratively and illustrated visually.

The final models will be presented with a discussion of their strengths and weaknesses. Furthermore, they will be considered in relation to practical application and their relevance for the organizational frameworks of future HaH solutions.”

3 Reviewer #3

RE Manuscript Number PONE-D-24-27648R2: The authors have presented a protocol for a scoping review on Hospital-at-Home structures and interventions for adults. Please find my detailed comments below.

3.1 I was unable to review the protocol on OSF. The authors should note that the OSF registration needs to outline the key information of the protocol clearly and transparently. Simply registering the title may not provide enough context for other researchers. You may refer to this for further information: https://osf.io/ym65x.

Response: Thankyou for bringing this to our attention. This is our first attempt with using the OSF platform and we were unaware that the upload had not worked as intended. We have now uploaded the 2 versions of the protocol as it has developed. The first version had a broad focus (did not include a Scandinavia block) and the second version that indicates our intention for a more narrow focus. Please see this link DOI: 10.17605/OSF.IO/WKE7X)for your information.

3.2 The abstract's transition from a general objective to focus on Scandinavian Hospital-at-Home is abrupt. The authors need to clearly state their focus in their objective.

Response: Thankyou for bringing the general, broad abstract to our attention. We agree the transition was very abrupt, too abrupt. We have adjusted the abstract, deleting and adding text and more clearly highlighting that the focus is within Scandinavia and our objectives for doing this. See highlighted text throughout the abstract (line 22-43, page 2). Reviewer 4 had comments of a similar nature (see 4.1)

“Objective

This proposed scoping review aims to map the components of Hospital-at-Home (HaH) organisational structures and interventions for the acute care of adults in Scandinavia.

Introduction

Healthcare systems face increased pressure due to ageing populations having multiple morbidities, leading to strains on hospitals. Hospitalization is associated with higher risks of adverse events, particularly among older individuals. The surge in hospital admissions for older adults necessitates prioritising resources and exploring alternative solutions. HaH has emerged as a promising intervention, showing favourable outcomes in patient satisfaction, stress reduction and quality of life. Recent health reform in Denmark has clearly indicated a clear political interest in HaH solutions. This review wishes to map Scandinavian HaH organizational collaboration, intervention set-up and patient demographics to assist Scandinavian healthcare administrators in understanding the current HaH models.

Inclusion criteria

Scandinavian studies based on adults with an acute illness requiring emergency care and initial contact with an emergency department…..

Methods

Peer-reviewed studies from the databases Medline, Embase and CENTRAL will be included without time limitation to present day and grey literature identifying information about HaH from Ministry of Health websites and clinicaltrial.gov will also be considered. Data extraction will include information on study and patient characteristics, organisational structures and intervention details as well as the measured effect outcomes. This protocol has been registered with Open Science Framework (DOI: 10.17605/OSF.IO/WKE7X).”

3.3 The methods of the abstract need to outline the date range of the study. Furthermore, focusing only on peer-reviewed articles is not in line with the methodology of a scoping review.

Response: Thank you for these reminders. As this is a protocol article, we are waiting for final feedback before setting the current date for the actual search. We have no plans to limit the search in relation to time. Databases will be searched from inception to the current date. We have adjusted this in the abstract as instructed. We have chosen to focus only on peer-reviewed articles from the health databases as we expect that it is improbable that other types of articles in these databases (eg.conference abstracts) include detailed organization descriptions. However, we agree that scoping reviews should examine grey literature, and we have chosen to look at health ministry websites in Scandinavian countries and the ClinicalTrials.gov database. We have included this information in the methods section in the abstract (line 40-41, page 2).

“Peer-reviewed studies from the databases Medline, Embase and CENTRAL will be included without time limitations and grey literature will be searched from Ministry of Health websites and ClinicalTrials.gov.”

3.4 The consistency and cohesion between the topics presented in the introduction needs to be revised. For example, after the paragraph discussing the origin of the research question, readers might expect a more direct continuation of that theme. However, the subsequent paragraphs introduce a new concept, which could disrupt the flow of the introduction.

Response: Thankyou for this constructive feedback. There have been a number of other reviewers that have commented on the flow of the introduction (See reviewer 6.4 and 8.1). We have made a number of changes to improve this focusing on the flow of topics. Please see line 45-79, page 3-4.

3.5 The review question needs to implement the PCC framework. At the time of writing this peer review, the review question is rather broad and difficult to navigate and understand. The authors may consider reformulating the research question in three distinct segments.

Response: We concur and using the PCC framework have reworded the research question. Please see line 81-83, page 4 and comments from reviewer #6.8

“How are HaH interventions for patients requiring emergency care organized in Scandinavia? What are the characteristics of the patients offered HaH care?”

3.6 I concur with the reviewers and editor that the focus on Scandinavian region may be too narrow for the purpose of a scoping review. However, since the authors are interested only in the Scandinavian region, I would recommend that they target the grey literature that are more relevant to that specific region. This source may provide useful databases that the author may consider to search and state in their protocol: https://www.hvl.no/en/library/find-databases-and-other-resources/.

Response: Thankyou for this suggestion. We referred the question of grey literature to our medical information specialist and decided, collaboratively, that ClincalTrials.gov and websites from health authorities was adequate in relation to our time and resources. Our stakeholders are primarily interested in published research in the field but could see the value of a limited grey literature search.

3.7 I would recommend the authors to summarize their data extraction in a table and present them to the readers. This will also save their time when extracting the results in the review process once you have preplanned data extraction table. The authors can then use the freed space in their protocol to articulate on how they will organize the data extraction. They can elaborate the process on data extraction, namely the minimum number of people involved, and how disagreements will be resolved.

Response: We appreciate your suggestion to include a preplanned data extraction table in our manuscript to improve clarity and efficiency in the review process. Reviewer 4.1 and 5.4 had similar concerns. As suggested, we have altered the data charting paragraph with the inclusion of Table 3 and Table 4 (see line 153-156, page 8-9) and expanded on the extraction process. We have also altered the data extraction heading to data charting process according to reviewer #4’s request (see 4.2 for changes in text and line 142-145, page 8 in the manuscript:

“The data charting process will be carried out using three researchers. One researcher (first author) will extract all data from all papers. This other two researchers will extract data blinded from the first author and any disagreements will be resolved by discussion.

3.8 Instead of the title 'Proposed data synthesis' the authors may consider 'Data analysis'.

Response: We have made this change as requested (see line 157, page 9).

3.9 I concur with the authors regarding risk of bias assessment being inappropriate in the context of a scoping review. Still, I would encourage the authors to include a statement in their protocol that they will not critically appraise the literature using any risk of bias assessment tools.

Response: Thankyou for your support with our assessment of relevance. We have included the following sentence under data analysis in the methods section (see line 172-173, page 9):

“We will not critically appraise the literature using risk of bias assessment tool, rather, all relevant studies will be included.”

3.10 Under the Discussion section, the authors have stated that by limiting the results to Scandinavian studies, the output will be homogenous. I would argue that some readers may misinterpret this from a statistical perspective, as in the homogeneity of data will allow the authors to conduct certain statistical analysis. I would recommend the authors using other terminologies to discuss this shortcoming and link this to what was expected from the Danish healthcare administrators.

Response: Thankyou for this comment. We agree that the word “homogenous” is often used with quantitative data in a statistical analysis. We have changed the word to “comparable” (line 198, page 10).

3.11 I would suggest the authors clearly state their dissemination plans once the review is over. This could include presentation of the findings in a peer-reviewed journal.

Response: Thanks for your suggestion. We have added the following to the manuscript (line 211-214, page 11):

“The review will be submitted for publication in a Scandinavian peer-reviewed journal and will be shared through institutional websites and LinkedIn. If possible, the preliminary results will be presented at an international conference”

3.12 PRISMA-P was not designed to report scoping review protocols. I would recommend the authors delete that in case of future submissions.

Response: Thanks for drawing this to our attention. We have been a little in doubt what reporting checklist to follow as there is no PRISMA extension for scoping review protocols as you have pointed out and PRISMA-ScR is used for the scoping review itself not the protocol article. PLOS one has a journal requirement that the PRISMA-P checklist is used and included.

4 Reviewer #4

Dear Respectable Authors

Thank you for considering a great area of research related to acute illness.

4.1 Your responses to previous reviewer is well but your manuscript needs some revisions as follows; - Abstract, methods section, please add some information regarding charting methods.

Response: Thanks for pointing out that information regarding charting methods should be added to our manuscript. Many of these issues have been address with other reviewers (see 3.7 and 5.4). Please see response to 3.2 (Abstract) and 3.7 (Methods and Data charting). See line 142-156, page 8-9 in the manuscript.

4.1 Based on the PRISMA-ScR, the eligibility criteria is the first step in reporting a scoping review. Please set this subheading as the first item in the methods section, after protocol and registration and then mention the search.

Response: Thankyou for identifying this improvement. PLOS one’s journal requirement request PRISMA-P checklist. As both PRISMA-P and PRISMA –ScR have eligibility criteria as the first step in the methods we have moved the heading as suggested. See line 92-97, page 4-5.

4.2 In scoping review we use the "data charting process" instead on "data extraction", then, it is better to use a separate subheading including "data charting process and data items" (Items 10 and 11 of PRISMA ScR).

Response: Thank you for your feedback. We acknowledge the useful distinction between "data extraction" and "data charting process" in scoping reviews and agree that it can be used in the manuscript.

5 Reviewer #5

This is a useful topic and appropriately justified. I understand the limit to the Scandinavian perspective if the research is being undertaken to satisfy particular stakeholders. I believe the methods that are selected are appropriate and have been adequately justified, particularly with regard to the time span and setting

Reponse: Thankyou for this feedback. Our stakeholders were solely interested in Scandinavian literature and although we explored broadening the search to start with – this was not possible logistically and did not fit with the aims of the stakeholders. Therefore the third block was added. We had understood that as a protocol article changes would be acceptable up until publication but can see that this has added confusion, unfortunately. See comments also from reviewer #3 and #6.

5.1 That said, more detail and transparency is needed for the literature search. It is not clear how exactly the peer review will be undertaken (PRESS peer review https://pubmed.ncbi.nlm.nih.gov/27005575/ is the most commonly used method) and how the authors commit to addressing any concerns.

Response: Thankyou for the opportunity to address this issue. The search strategy for Embase was peer-reviewed by a information specialist specialising in medicine and health. This

---

## [Decision Letter · Decision Letter 3]

PONE-D-24-27648R3Mapping the organisational and interventional framework for patients admitted to Hospital-at-Home for acute illness in Scandinavia – a scoping review protocolPLOS ONE

Dear Dr. Hansen,

Thank you for submitting your manuscript to PLOS ONE. After careful consideration, we feel that it has merit but does not fully meet PLOS ONE’s publication criteria as it currently stands. Therefore, we invite you to submit a revised version of the manuscript that addresses the points raised during the review process.

We look forward to receiving your revised manuscript.

Kind regards,

Mickael Essouma, M. D.

Academic Editor

PLOS ONE

**Additional Editor Comments:**

The authors have partly improved the manuscript. However, I mentioned in the last round of review that the abstract should not have sub-sections. The authors can verify that information in contemporary PLOS One articles online. However, the abstract should be written as usual for protocol articles, i.e., an initial statement on the background knowledge (introduction), followed by a statement of the aim/objective, a statement of the review methods, and finally a statement on ethics and planned mode of review dissemination. Based on the current abstract, the authors could rephrase the abstract like this (the statement on ethics and dissemination will need to be added at the end of this text):

"Hospitals in Scandinavia increasingly face an enormous pressure to manage acute emergencies in adults affected by multimorbid disabling conditions and therefore at risk of developing adverse hospitalisation outcomes such as nosocomial diseases. In this context, there is a growing political interest in the region to develop alternative models of acute emergency care such as the Hospital-at-Home (HaH), all the more so as adult patients themselves are showing a pronounced interest in HaH". We are therefore planning a scoping review to map the HaH components and interventions delivered to patients when managing acute emergencies in adults with that hospitalization model. We will search the databases MEDLINE, Embase and CENTRAL (Cochrane Central Register of Controlled trials) to include peer-reviewed articles of studies on adults admitted to HaH for acute emergency care within 24 hours of getting in touch with either an adult emergency department, an out-of-hours doctor, or a general practitioner. No limitation will be placed on the search period. The electronic search will be supplemented by a grey literature search of ClinicalTrials.gov and of the archives of Ministries of Health of the Scandinavian region. The information recorded during the data extraction process will include mainly the organisational structures of HaH, interventions delivered to adults with acute emergencies in HaH, and the characteristics of those patients. Data will be synthesized narratively. This protocol has been registered with Open Science Framework (DOI: 10.17605/OSF.IO/WKE7X). Mapping data on HaH for the care of adult acute emergencies in Scandinavia will help provide Scandinavian healthcare stakeholders with an overview of the strengths and weaknesses of different existing HaH models so that they can they can integrate that knowledge to make context-specific recommendations about and subsequently formally implement the management of adult acute emergencies in HaH in the region." The abstract should be followed by keywords which are missing in the current manuscript. In addition, Consider addressing my comments (of the first round of review) for the other sections of the manuscript. Those comments are available in the document PONE-D-24-27648_R2_Mickael Essouma comments.pdf attached to this decision letter

Mickael Essouma

Reviewers' comments:

Reviewer's Responses to Questions

**Comments to the Author**

1. Does the manuscript provide a valid rationale for the proposed study, with clearly identified and justified research questions?

Reviewer #3: Partly

Reviewer #5: Yes

Reviewer #6: Yes

Reviewer #8: Yes

2. Is the protocol technically sound and planned in a manner that will lead to a meaningful outcome and allow testing the stated hypotheses?

Reviewer #3: Partly

Reviewer #5: Partly

Reviewer #6: Partly

Reviewer #8: Yes

3. Is the methodology feasible and described in sufficient detail to allow the work to be replicable?

Reviewer #3: Yes

Reviewer #5: Yes

Reviewer #6: Yes

Reviewer #8: Yes

4. Have the authors described where all data underlying the findings will be made available when the study is complete?

Reviewer #3: Yes

Reviewer #5: Yes

Reviewer #6: Yes

Reviewer #8: Yes

5. Is the manuscript presented in an intelligible fashion and written in standard English?

Reviewer #3: Yes

Reviewer #5: Yes

Reviewer #6: Yes

Reviewer #8: Yes

6. Review Comments to the Author

You may also provide optional suggestions and comments to authors that they might find helpful in planning their study.

Reviewer #3: I would like to thank the authors for submitting a revised version of their manuscript. Below are my comments for further improvements:

1) When registering the study on OSF, the authors should complete the registration form directly on the platform rather than uploading a separate protocol document. For reference, see this example registration: https://doi.org/10.17605/OSF.IO/RQKTP

2) While reviewing abstracts of scoping reviews published in PLOS One, I noted that some use an unstructured format. The authors may consider reordering their abstract to begin with the introduction followed by the objectives to improve logical flow.

3) The two review questions would benefit from being numbered (e.g., "1)", "2)") rather than presented in a single paragraph to enhance readability.

4) The exclusion of grey literature remains problematic. If stakeholder preferences preclude its inclusion, the authors should either:

a) Transition to a systematic review with a risk of bias assessment and GRADE analysis, or

b) Reframe the study as a narrative literature review.

Scoping reviews require exhaustive searches; limiting sources may compromise the methodology’s validity.

Reviewer #5: Thank you for the updated and improved version of this manuscript. I have limited my comments to my own concerns rather than issues raised by other reviewers. I have answered "yes" to #3 and #4, above, but qualify this as "partly", as per my comments, below.

There is a mixture of future and past tense when describing the search strategy - “Will be reviewed” but then the paragraph notes that the the preliminary strategy was sent for review. The strategy seems to have been developed collaboratively rather than one information specialist responding to the PRESS checklist. There are processes to be followed when using PRESS (e.g., any required revisions necessitate a second formal review/PRESS). A reference to the PRESS checklist needs to be inserted and it would be helpful to include the completed PRESS so that it is clear how any issues were addressed.

I don’t think necessary to note that publications will be restricted to peer-review studies since this is implied through the choice of bibliographic databases.

The strategy provided actually uses MeSH headings rather than Emtree, so I suspect the preliminary strategy was developed in MEDLINE, not Embase, although this is not obvious without including the banner from the executed strategy. Only one of the headings uses an exclusively Emtree term, “Scandinavia”, as the MeSH is “Scandinavian and Nordic Countries” (Note: “Greenland” is not covered under this term and would require a separate entry in MEDLINE).

There are still deficiencies in the search strategy that should be addressed (several lines, e.g., 9, 10, 11, 19, 20, 21, 25, 32, 37, 41, 42, 46, should include both singular and plural options; some terms may appear without hyphens, e.g., homecare, homebased, hospitalbased, sameday; the regional terms should include Icelandic, Icelander*, Greenlander*, Nordic country; a mandatory wildcard should be used rather than an optional one for e.g., hospitali#ation.

The methods also state the grey literature from ClinicalTrials.gov will be imported into Covidence – this is redundant with the study selection that follows so I suggest removal.

In discussing the limitations of the narrow grey literature, I would suggest using the term “missed” rather than “excluded”, which has a different meaning. Similarly, the note that the stakeholders had limited interest in non-peer reviewed documents is not connected to the restriction to the Scandinavian region, but rather to the grey literature. Note simply that the regional focus was considered central to the stakeholders’ interests, so was deemed necessary to e.g., avoid inflating volume. A targeted grey literature search was adopted for the same reasons.

Some small grammatical notes:

Need hyphenation:

Table 1: out-of-hours doctor

Database-specific subject headings

Thank you for the opportunity to review this updated version.

Reviewer #6: Dear authors,

I have received the revised version of your manuscript. Below are my observations.

First, I would like to take a moment to congratulate you on your work and to thank you for the effort dedicated to revising this protocol. Science is built through these spaces for discussion, and I appreciate your openness to critically engage with your own work.

Regarding the review of your manuscript, although most of the comments (both mine and those of the other reviewers) have been addressed, there are still some fundamental issues that should be revised before considering the work ready for dissemination. My final comments are as follows:

Abstract:

• If the abstract sections are determined by the journal’s structure, I suggest revisiting the content of each section. For example, in the methods section, it is not mentioned that this is a scoping review nor that it will follow the methodology proposed by JBI. This would be a good starting point. If there is no indication that a structured abstract is required, I suggest reorganizing the sections. Possible structure for consideration: Introduction (Objective), Methods, Expected Results (Implication of Results). However, you may reformulate it in the way that seems most appropriate, aiming to provide as much detail as possible within the word limit.

Introduction:

• I recommend providing more references to support the following statement: “Hospital-at-Home (HaH) has shown to reduce adverse events and distress 48 associated with hospitalization of adults and ease the burden on hospitals [4].” Additionally, the definition of HaH is only provided in the inclusion criteria section, which could result in readers unfamiliar with the term not understanding it upon first encounter. I recommend logically incorporating the definition of HaH within the introduction, and then referring back to it in the criteria section to avoid repetition (this strategy generally works well).

• Regarding the statement: “Healthcare systems are under pressure with an unprecedented number of people reaching retirement and living longer with multiple illnesses,” is this really the core problem? As phrased, it seems to suggest that living longer is the problem, when in reality, the issue is structural (as you note later). I suggest rethinking this idea.

• "One promising intervention is HaH." In line 54, this idea is introduced abruptly. Please review its integration into the text's logical flow.

• Line 61: A period is missing.

• Line 64: “Cochrane review.” Referring to the review this way may not be appropriate, as it appeals to the methodology rather than the study type, and creates an unnecessary distinction among reviews (not advisable). I suggest mentioning the authors or simply stating “a systematic review.”

• If there are no previous reviews in the Scandinavian context, it would be helpful to explicitly mention this. JBI recommends including the absence of prior reviews (reviews, not individual studies).

General comment:

The logical and syntactic structure of the introduction remains unclear and may now be the main weakness of the manuscript. I suggest revising it. Avoid offering statements without proper contextualization. Short sentences can fragment the comprehension of meanings. Consider adopting a logical structure (deductive or inductive) and writing accordingly.

Methods:

• Line 88: The "scoping review" methodology is not the JBI manual. The manual contains various methodologies. I suggest revising to: “methodology proposed by the Joanna Briggs Institute (JBI).”

• Line 91: “This protocol has been uploaded to Open Science Framework (DOI: 10.17605/OSF.IO/WKE7X).” Protocols are not “uploaded”; they are registered. This is correctly referenced in the abstract; please revise here.

Search Strategy:

• I was struck by the use of only “CENTRAL (Cochrane).” Will you not review systematic reviews? If not, it is not methodologically incorrect, but it should be clearly stated and justified. Additionally, I suggest providing the definition of CENTRAL (Cochrane Central Register of Controlled Trials).

• Line 112: “…controlled vocabulary thesaurus.” The expression "controlled vocabulary thesaurus" is redundant, as a thesaurus inherently represents a controlled vocabulary. It would be clearer to refer simply to "controlled vocabulary" or "thesaurus," depending on the intended emphasis.

• General comment: Review the “search strategy” section according to JBI methodology. This will help you report the procedures sequentially and logically.

• Line 114: Review the standardized use of acronyms (HaH).

• Line 116: If you have decided to include grey literature, it is inappropriate to state: “Publication type will be restricted to peer-reviewed.” I suggest revising this here and in other parts of the text where it appears.

• Line 118: “observational and descriptive.” Are there differences? What exactly do you mean by descriptive? This would also be a good point to clarify the comment regarding systematic reviews.

• Lines 119–120: “Forward citation searches of included studies will be performed using the Web of Science database to identify any additional literature.” This statement is unclear. To which process are you specifically referring? Reverse citation searches? That is, searching references of potentially relevant articles? Or searching Web of Science more broadly? If it is the latter, WoS should be listed as a database in its own right (recommended). If not, please clarify.

• Lines 121–124: “Grey literature from ClinicalTrials.gov will be imported into Covidence and screened with double-blinding. One author will screen any relevant website information and if considered relevant the information will be screened by a second author.” This paragraph should be included at the end of the “study selection” section.

• Lines 128–131: “The scoping review methodology was chosen for this review because stakeholders wanted to map the existing literature published within Scandinavia and identify gaps in HaH research. A scoping review was also deemed relevant when the aim was to compare organisational models and provide a broader, more descriptive synthesis of the research landscape for acute HaH patients in Scandinavia.” This paragraph should be removed from this section and logically integrated into the introduction as part of the review’s justification.

• Line 140: Review according to the comment from “Line 88.”

• Lines 139–140: In the previous version, my comments suggested adding “agreement between reviewers during study selection.” Although the authors provided a response, it is not fully satisfactory according to the methodology. I will cite the section from the JBI manual I am referring to:

We recommend some pilot testing of source selectors prior to embarking on source selection across a team. This will allow the review group to refine their guidance or source selection tool (if one is being used). One framework for pilot testing is described below: - Random sample of 25 titles/abstracts is selected - The entire team screens these using the eligibility criteria and definitions/elaboration document - Team meets to discuss discrepancies and make modifications to the eligibility criteria and definitions/elaboration document - Team only starts screening when 75% (or greater) agreement is achieved.

Data Charting Process:

Now the process is clearer. As a final recommendation, I suggest adding that the variables to be extracted may be modified according to the findings. Remember that this is a protocol and it is always subject to change based on new insights.

Data Analysis:

There are fragments that refer again to the data extraction process. Please review for clarity and avoid repetition.

Discussion:

Ensure that the discussion does not simply extend the introduction. I suggest describing how the results will impact the various areas related to this investigation.

Line 184: Isn’t offered a definition to “EDs”

Final Comment:

The manuscript is now clearer. I encourage you to critically engage with the observations made, not treating them as rigid recommendations, but rather as perspectives that may strengthen your internal debate. Since this is a protocol and does not yet present results, you should take advantage of the opportunity to write with sufficient detail about all planned procedures.

Reviewer #8: The article has undergone a remarkable revision, and I deeply value the authors' arduous efforts. Nevertheless, there remain several minor suggestions that require attention and resolution:

1. Abstract section: The content is relatively concise and covers the research purpose. However, it would be beneficial to appropriately add a brief description of the research methods and expected results, enabling readers to have a more comprehensive and rapid understanding of the core key points of the research.

2. Research selection: The process of literature screening using the Covidence platform is clear. However, there is no explanation regarding possible special situations during the screening process, such as how to handle documents in different languages and how to deal with the situation when the quality of the literature varies. Relevant content can be supplemented.

3. Data analysis: The proposed data analysis method is relatively reasonable. However, when elaborating on how to organize the model according to data classification and iterative development, there is a lack of specific operation steps and examples. It can be further refined to make the research method more operational.

7. PLOS authors have the option to publish the peer review history of their article (what does this mean? ). If published, this will include your full peer review and any attached files.

**Do you want your identity to be public for this peer review?** For information about this choice, including consent withdrawal, please see our Privacy Policy .

Reviewer #3: No

Reviewer #5: No

Reviewer #6: **Yes: ** Romel Jonathan Velasco Yanez

Reviewer #8: No

---

## [Author Response · Author response to Decision Letter 4]

10 Jun 2025

PONE-D-24-27648R3

Mapping the organisational and interventional framework for patients admitted to Hospital-at-Home for acute illness in Scandinavia – a scoping review protocol

PLOS ONE

Response:

We sincerely thank you and the reviewers for the detailed and constructive feedback provided throughout this review process. We appreciate the considerable time and effort invested in the careful reading of our manuscript and the thoughtful comments offered across several rounds of revision. In response, we have made the suggested adjustments and have worked to improve the clarity, coherence, and overall quality of the manuscript accordingly.

Response to feedback from the editor:

Our abstract is adjusted as suggested removing subheading and adding keywords and ethical considerations. We have added a sentence in the method sections as you suggested in your response to the first round of review (line 94-98, page 5-6).

“The development of this review protocol was undertaken in collaboration with key stakeholders who expressed a specific interest in synthesising evidence related to Hospital at Home services within the Scandinavian context. In response to their input, the original search strategy was refined to ensure greater relevance to the geographical and contextual focus of the inquiry. The current strategy reflects this adjustment, incorporating terminology and sources tailored to the Scandinavian healthcare setting.”

This is the abstract that you suggested, similar to reviewer #3, #6 and #8´s comments (line 22-45, page 2):

"Hospitals in Scandinavia increasingly face an enormous pressure to manage acute emergencies in adults affected by multimorbid disabling conditions and therefore at risk of developing adverse hospitalisation outcomes such as nosocomial diseases. In this context, there is a growing political interest in the region to develop alternative models of acute emergency care such as the Hospital-at-Home (HaH), all the more so as adult patients themselves are showing a pronounced interest in HaH. We are therefore planning a scoping review to map the HaH components and interventions delivered to patients when managing acute emergencies in adults with that hospitalization model. We will search the databases MEDLINE, Embase and CENTRAL (Cochrane Central Register of Controlled trials) to include peer-reviewed articles of studies on adults admitted to HaH for acute emergency care within 24 hours of getting in touch with either an adult emergency department, an out-of-hours doctor, or a general practitioner. No limitation will be placed on the search period. The electronic search will be supplemented by a grey literature search of ClinicalTrials.gov and of the archives of Ministries of Health of the Scandinavian region. The information recorded during the data extraction process will include mainly the organisational structures of HaH, interventions delivered to adults with acute emergencies in HaH, and the characteristics of those patients. Data will be synthesized narratively. This protocol has been registered with Open Science Framework (DOI: 10.17605/OSF.IO/WKE7X). Mapping data on HaH for the care of adult acute emergencies in Scandinavia will help provide Scandinavian healthcare stakeholders with an overview of the strengths and weaknesses of different existing HaH models so that they can they can integrate that knowledge to make context-specific recommendations about and subsequently formally implement the management of adult acute emergencies in HaH in the region.

Ethical approval was not required as the study does not involve human participants. Findings will be submitted for publication in a peer-reviewed Scandinavian journal and disseminated through institutional websites and LinkedIn. Results will be presented at an international conference."

Reviewer #3:

I would like to thank the authors for submitting a revised version of their manuscript. Below are my comments for further improvements:

1) When registering the study on OSF, the authors should complete the registration form directly on the platform rather than uploading a separate protocol document. For reference, see this example registration: https://doi.org/10.17605/OSF.IO/RQKTP

Response: Thank you for your comment. We have now corrected this by completing the registration form directly on the OSF platform, as recommended.

2) While reviewing abstracts of scoping reviews published in PLOS One, I noted that some use an unstructured format. The authors may consider reordering their abstract to begin with the introduction followed by the objectives to improve logical flow.

Response: Thank you for your helpful comment. In line with the editor’s comment, we have revised the abstract to follow an unstructured format. We have used the version suggested by the editor. See line 22-45, page 2.

3) The two review questions would benefit from being numbered (e.g., "1)", "2)") rather than presented in a single paragraph to enhance readability.

Response: Thank you for your helpful suggestion. We have revised the manuscript by numbering the two review questions “1)” and “2)” instead of presenting them in a single paragraph, to enhance clarity and readability. See line 83-85, page 5.

4) The exclusion of grey literature remains problematic. If stakeholder preferences preclude its inclusion, the authors should either:

a) Transition to a systematic review with a risk of bias assessment and GRADE analysis, or

b) Reframe the study as a narrative literature review.

Scoping reviews require exhaustive searches; limiting sources may compromise the methodology’s validity.

Response: I think there has been a misunderstanding. We did not exclude grey literature. See line 34-35, page 2 and line 129-130, page 8.

Reviewer #5:

Thank you for the updated and improved version of this manuscript. I have limited my comments to my own concerns rather than issues raised by other reviewers. I have answered "yes" to #3 and #4, above, but qualify this as "partly", as per my comments, below.

There is a mixture of future and past tense when describing the search strategy - “Will be reviewed” but then the paragraph notes that the the preliminary strategy was sent for review. The strategy seems to have been developed collaboratively rather than one information specialist responding to the PRESS checklist. There are processes to be followed when using PRESS (e.g., any required revisions necessitate a second formal review/PRESS). A reference to the PRESS checklist needs to be inserted and it would be helpful to include the completed PRESS so that it is clear how any issues were addressed.

I don’t think necessary to note that publications will be restricted to peer-review studies since this is implied through the choice of bibliographic databases.

Response: We appreciate your comment regarding the phrase “restricted to peer-reviewed studies.” However, we respectfully maintain this wording to clarify our inclusion criteria from the databases (see line 123-124, page 8). In databases such as Embase, a substantial proportion of records consist of conference abstracts, preprints, and other forms of literature that have not undergone formal peer review. We hope we have clarified this now.

The strategy provided actually uses MeSH headings rather than Emtree, so I suspect the preliminary strategy was developed in MEDLINE, not Embase, although this is not obvious without including the banner from the executed strategy. Only one of the headings uses an exclusively Emtree term, “Scandinavia”, as the MeSH is “Scandinavian and Nordic Countries” (Note: “Greenland” is not covered under this term and would require a separate entry in MEDLINE).

There are still deficiencies in the search strategy that should be addressed (several lines, e.g., 9, 10, 11, 19, 20, 21, 25, 32, 37, 41, 42, 46, should include both singular and plural options; some terms may appear without hyphens, e.g., homecare, homebased, hospitalbased, sameday; the regional terms should include Icelandic, Icelander*, Greenlander*, Nordic country; a mandatory wildcard should be used rather than an optional one for e.g., hospitali#ation.

Response:

Thank you very much for your insightful comments regarding our search strategy and the distinctions between MeSH and Emtree terms in MEDLINE and EMBASE. After reflecting on your feedback, we carefully re-examined our search string.

You were correct that many of the MeSH terms originally included were specific to the MEDLINE database. We have now adjusted the search accordingly, including correcting wildcard use as suggested and broadening the country-specific block to ensure better coverage.

During this process, we discussed the difference between the .kf and .kw fields. Based on this, we have updated our search to use .kw to better reflect author-chosen keywords.

We also considered your helpful comments about wildcard use. Specifically, we explored whether the optional wildcard symbol (#) might help address variation in hyphenated terms such as home-based care. However, we found that the # symbol in Ovid substitutes only for a single optional character and is therefore not suitable for capturing common structural variations like home-based care, home based care, and homebased care. To ensure a more comprehensive and accurate search, we chose to explicitly include all three forms in our search string.

We sincerely thank you for the opportunity to revisit and improve our strategy. Your detailed and constructive feedback has significantly strengthened our search and contributed to our learning in the process. See Table 2, page 7-8.

Lastly, we have included a reference to the PRESS (Peer Review of Electronic Search Strategies) guideline to acknowledge best practice, although we have chosen not to include the full checklist, as its use is optional (line 112-113, page 7).

Thank you again for your valuable input.

The methods also state the grey literature from ClinicalTrials.gov will be imported into Covidence – this is redundant with the study selection that follows so I suggest removal.

Response: Although literature from Clinical Trials is grey literature, our stakeholders are specifically interested in any trials in the region that may be being conducted on this area.

In discussing the limitations of the narrow grey literature, I would suggest using the term “missed” rather than “excluded”, which has a different meaning. Similarly, the note that the stakeholders had limited interest in non-peer reviewed documents is not connected to the restriction to the Scandinavian region, but rather to the grey literature. Note simply that the regional focus was considered central to the stakeholders’ interests, so was deemed necessary to e.g., avoid inflating volume. A targeted grey literature search was adopted for the same reasons.

Response: Thanks for these comments. We have adjusted the manuscript accordingly - See line 127, page 8

Some small grammatical notes:

Need hyphenation:

Table 1: out-of-hours doctor

Database-specific subject headings

Response: Thanks. We appreciate you pointing out the grammatical issues. We have added the appropriate hyphenation in Table 1, correcting “out of hours doctor” to “out-of-hours doctor,” (line 104, page 6). “Database specific subject headings” has been deleted.

Thank you for the opportunity to review this updated version.

Reviewer #6:

Dear authors,

I have received the revised version of your manuscript. Below are my observations.

First, I would like to take a moment to congratulate you on your work and to thank you for the effort dedicated to revising this protocol. Science is built through these spaces for discussion, and I appreciate your openness to critically engage with your own work.

Regarding the review of your manuscript, although most of the comments (both mine and those of the other reviewers) have been addressed, there are still some fundamental issues that should be revised before considering the work ready for dissemination. My final comments are as follows:

Abstract:

• If the abstract sections are determined by the journal’s structure, I suggest revisiting the content of each section. For example, in the methods section, it is not mentioned that this is a scoping review nor that it will follow the methodology proposed by JBI. This would be a good starting point. If there is no indication that a structured abstract is required, I suggest reorganizing the sections. Possible structure for consideration: Introduction (Objective), Methods, Expected Results (Implication of Results). However, you may reformulate it in the way that seems most appropriate, aiming to provide as much detail as possible within the word limit.

Response: Thank you for your helpful suggestion. The abstract has been revised and reformulated to improve clarity and coherence, as also recommended by Editor and Reviewer #3 and #8. We have explicitly stated that the study is a scoping review and that it follows the methodology proposed by the Joanna Briggs Institute (JBI). These revisions have been made to ensure that the abstract provides a clear and structured overview of the study design and approach, in line with journal expectations. See line 27-28, page 2 and line 88-90, page 5.

Introduction:

• I recommend providing more references to support the following statement: “Hospital-at-Home (HaH) has shown to reduce adverse events and distress 48 associated with hospitalization of adults and ease the burden on hospitals [4].” Additionally, the definition of HaH is only provided in the inclusion criteria section, which could result in readers unfamiliar with the term not understanding it upon first encounter. I recommend logically incorporating the definition of HaH within the introduction, and then referring back to it in the criteria section to avoid repetition (this strategy generally works well).

• Regarding the statement: “Healthcare systems are under pressure with an unprecedented number of people reaching retirement and living longer with multiple illnesses,” is this really the core problem? As phrased, it seems to suggest that living longer is the problem, when in reality, the issue is structural (as you note later). I suggest rethinking this idea.

• "One promising intervention is HaH." In line 54, this idea is introduced abruptly. Please review its integration into the text's logical flow.

• Line 61: A period is missing.

• Line 64: “Cochrane review.” Referring to the review this way may not be appropriate, as it appeals to the methodology rather than the study type, and creates an unnecessary distinction among reviews (not advisable). I suggest mentioning the authors or simply stating “a systematic review.”

• If there are no previous reviews in the Scandinavian context, it would be helpful to explicitly mention this. JBI recommends including the absence of prior reviews (reviews, not individual studies).

Response: Thank you for your constructive and detailed feedback. We have carefully addressed all your suggestions:

• We have added additional references to support the statement regarding the benefits of Hospital-at-Home (HaH), thereby strengthening the evidence base (line 52-54, page 4).

• The definition of HaH has been added to the introduction for clarity and improved logical flow (line 55-58, page 4).

• The sentence about healthcare systems under pressure has been revised to clarify that the challenge lies in structural strain rather than longevity itself (line 52-55, page 4).

• The sentence “One promising intervention is HaH” has been integrated into the surrounding context, ensuring smoother narrative flow (line 51-52, page 4).

• The missing period has been inserted.

• We have replaced the term “Cochrane review” with “systematic review,” as advised, to avoid unnecessary methodological emphasis (line 66, page 4).

• Finally, we have explicitly noted the absence of previous reviews in the Scandinavian context, in accordance with JBI recommendations (line 72-73, page 4-5).

General comment:

The logical and syntactic structure of the introduction remains unclear and may now be the main weakness of the manuscript. I suggest revising it. Avoid offering statements without proper contextualization. Short sentences can fragment the comprehen

---

## [Decision Letter · Decision Letter 4]

PONE-D-24-27648R4Mapping the organisational and interventional framework for patients admitted to Hospital-at-Home for acute illness in Scandinavia – a scoping review protocolPLOS ONE

Dear Dr. Hansen,

Thank you for submitting your manuscript to PLOS ONE. After careful consideration, we feel that it has merit but does not fully meet PLOS ONE’s publication criteria as it currently stands. Therefore, we invite you to submit a revised version of the manuscript that addresses the points raised during the review process.

We look forward to receiving your revised manuscript.

Kind regards,

Mickael Essouma, M. D.

Academic Editor

PLOS ONE

Journal Requirements:

Additional Editor Comments:

The authors have improved their manuscript and the reviewer has recommended acceptation of the manuscript for publication. However, there is a need for more precise organisation of the text, justifying the decision of minor revision before the manuscript can be accepted for publication.

Point-by-point comments below

1. Line 83: consider writing "Review questions" rather than "Review question".

2. Inclusion and exclusion criteria sub-section of the Materials and Methods section: consider citing inclusion and exclusion criteria other than those derived from the PCC typology as well after table 2. Notably, the sentences "Any relevant studies...our review questions." (lines 122-126) and "Authors will be contacted...as the one HaH concept." (lines 130-132) should be inserted after table 2 given that they are highlighting inclusion and exclusion criteria (type of publication to be excluded, study designs to be included, and potentially exclusion of records for which full-texts will be unavailable even after contacting the author, at least for records retrieved from Medline and Embase). The authors should also state why they place a limitation on peer reviewed articles only for Medline and Embase databases, and not for the other electronic databases that will be searched. Even after reading the last sentence in lines 131 and 132, it is unclear to me how they will proceed for the inclusion of salami-sliced and duplicate articles. Consider also specifying that in the sub-section Inclusion and exclusion criteria. Information about the types of publications (editorials without data?) that will be excluded is also warranted.

3. Data charting process sub-section

Consider subdividing the variables into three categories throughout the text and in the corresponding table: study characteristics, participant characteristics and main review outcomes, i.e., interventions and organisational structures of HaH. I suggest deleting tables 3 and 4 from the main manuscript given that they are just templates of the data extraction sheet. You could merge them in a single table titled "Template of data extraction sheet that will be used" in a way that information about the study characteristics is provided in arrows above, while the overview of interventions, organisational structures and patient outcomes are provided in the arrows below. You would then send that table to the appendix. Regarding the study characteristics to be mentioned in the template of data extraction sheet/preconceived data extraction sheet, consider adding the following variables: locality where the study was conducted (urban versus rural versus aemi-urban), sampling technique used, overall sample size and sample size of participants admitted to HaH, iinformation about the economic status of participants, and information about the comparator population of any.

4.Data analysis sub-section.It appears nowhere that you will conduct a narrative synthesis as expected in a scoping review protocol.

5. As proposed in a previous round of revision, consider inserting a "Patient and public involvement" sub-section after the Data analysis sub-section, stating whether members of the public, people with lived disease experience of acute illness management in HaH. If you do not plan to include them, then simply write "None" under the "Patient and public involvement" sub-title. As highlighted in the previous round of revision, consider also inserting a "Protocol deviation"

sub-section just before the sub-section "Patient and public involvement" to specify how you would deal with any protocol modification after the publication of this manuscript. It is important to remind that protocol deviation is possible and this has been done since this protocol article has entered the review process. So, I find it appropriate to plan ahead.

6. Other comments

References. I would delete reference 14 and use reference 23 for backing up the comment that was backed up by ref 14. Reference 24 is the same as reference 16.

Line 149: consider replacing "using" by "by".

Line 151: consider writing: "in a tabular form".

Line 199: consider replacing "in the home" by "at home".

Consider adding conflicts of interests and data availability statements before the reference section.

It is unclear to me whether you implemented Reviewer 3' s comment about the protocol registration on the OSF. I am unable to access the review protocol using the doi number provided in lines 38 and 92. Consider addressing this issue.

Available online on 26/06/2025

Mickael Essouma

Reviewers' comments:

Reviewer's Responses to Questions

**Comments to the Author**

1. Does the manuscript provide a valid rationale for the proposed study, with clearly identified and justified research questions?

Reviewer #5: Yes

2. Is the protocol technically sound and planned in a manner that will lead to a meaningful outcome and allow testing the stated hypotheses?

Reviewer #5: Yes

3. Is the methodology feasible and described in sufficient detail to allow the work to be replicable?

Reviewer #5: Yes

4. Have the authors described where all data underlying the findings will be made available when the study is complete?

Reviewer #5: Yes

5. Is the manuscript presented in an intelligible fashion and written in standard English?

Reviewer #5: Yes

6. Review Comments to the Author

You may also provide optional suggestions and comments to authors that they might find helpful in planning their study.

Reviewer #5: Thank you for the opportunity to review the revised manuscript. I have limited my comments to my own concerns but note that other reviewers have generally responded positively to the most recent revisions which they requested. I have recommended "accept" but there are a few very minor adjustments that could be made pertaining to the search methods and strategy.

I disagree with the comment "We will search CENTRAL rather than the full Cochrane Library as our interest is limited to peer-reviewed original studies which CENTRAL specifically indexes". CENTRAL includes preprints and records from clinical trial registries, which would not be peer reviewed.

Within the search strategy, the following is noted:

exp Hospital to Home Transition/ (there are no subtrees so shouldn't explode - no effect on search)

By exploding "Scandinavian and Nordic Countries" - all but "Greenland" are already covered so listing them as separate MeSH is redundant. The only one required is "Greenland/"

Suggest adding truncation to pick up plurals for Norwegian*, Northern Europe*, and consider adding Swede, Swedes

7. PLOS authors have the option to publish the peer review history of their article (what does this mean? ). If published, this will include your full peer review and any attached files.

**Do you want your identity to be public for this peer review?** For information about this choice, including consent withdrawal, please see our Privacy Policy .

Reviewer #5: No

---

## [Author Response · Author response to Decision Letter 5]

3 Jul 2025

Response to Reviewers

Dear Editor and Reviewer#5,

Thank you for the opportunity to further refine and improve our protocol. We truly appreciate the time and thoughtful feedback provided, which we believe will lead to a stronger and more rigorous final review.

We have carefully addressed the concerns raised in the latest round of review, as outlined below:

Journal Requirements:

Additional Editor Comments:

The authors have improved their manuscript and the reviewer has recommended acceptation of the manuscript for publication. However, there is a need for more precise organisation of the text, justifying the decision of minor revision before the manuscript can be accepted for publication.

1. Line 83: consider writing "Review questions" rather than "Review question".

Response: Completed

2. Inclusion and exclusion criteria sub-section of the Materials and Methods section: consider citing inclusion and exclusion criteria other than those derived from the PCC typology as well after table 2. Notably, the sentences "Any relevant studies...our review questions." (lines 122-126) and "Authors will be contacted...as the one HaH concept." (lines 130-132) should be inserted after table 2 given that they are highlighting inclusion and exclusion criteria (type of publication to be excluded, study designs to be included, and potentially exclusion of records for which full-texts will be unavailable even after contacting the author, at least for records retrieved from Medline and Embase). The authors should also state why they place a limitation on peer reviewed articles only for Medline and Embase databases, and not for the other electronic databases that will be searched. Even after reading the last sentence in lines 131 and 132, it is unclear to me how they will proceed for the inclusion of salami-sliced and duplicate articles. Consider also specifying that in the sub-section Inclusion and exclusion criteria. Information about the types of publications (editorials without data?) that will be excluded is also warranted.

Response:

We understand that you would like all information regarding the inclusion and exclusion criteria to be consolidated in one location within the manuscript. In your comment, you referred to moving text "after Table 2," but we assume this may have been a typographical error and that you intended to refer to Table 1. In response, we have now summarized all relevant details about the inclusion and exclusion criteria in the paragraph accompanying Table 1. Our selection focused on original research identified through MEDLINE and Embase, to ensure that included studies provided sufficient detail—particularly on the organizational structures of HaH services. As a result, we excluded books, editorials, conference abstracts, duplicates and salami-sliced articles.

However, in recognition of stakeholder interest in a broader overview, we also included select relevant non–peer-reviewed sources. This rationale and the full criteria are now clearly presented on line 106-114, page 6-7.

3. Data charting process sub-section

Consider subdividing the variables into three categories throughout the text and in the corresponding table: study characteristics, participant characteristics and main review outcomes, i.e., interventions and organisational structures of HaH. I suggest deleting tables 3 and 4 from the main manuscript given that they are just templates of the data extraction sheet. You could merge them in a single table titled "Template of data extraction sheet that will be used" in a way that information about the study characteristics is provided in arrows above, while the overview of interventions, organisational structures and patient outcomes are provided in the arrows below. You would then send that table to the appendix. Regarding the study characteristics to be mentioned in the template of data extraction sheet/preconceived data extraction sheet, consider adding the following variables: locality where the study was conducted (urban versus rural versus aemi-urban), sampling technique used, overall sample size and sample size of participants admitted to HaH, iinformation about the economic status of participants, and information about the comparator population of any.

Response:

Thank you for your helpful and constructive suggestions regarding the “Data charting process” section. As requested, we have revised the manuscript to reflect your recommendations:

• Variables are now grouped into three categories: study characteristics, participant characteristics, and main review outcomes (interventions and organisational structures of HaH) and we have revised it consistently throughout the text. See line 36, page 2 and line 159, page 10

• Tables 3 and 4 have been removed from the main text. We have merged them into a single table titled “Template of data extraction sheet that will be used”, now placed in the S1 Table.

• Additional study characteristics have been added to the template (S1 Table), including: study locality (urban/semi-urban/rural), sampling technique, total and HaH sample sizes and comparator group details.

4. Data analysis sub-section. It appears nowhere that you will conduct a narrative synthesis as expected in a scoping review protocol.

Response: Thank you for your observation. We have now included on line 37, page 2 and line 165 and line 173, page 10 that the data will be synthesized narratively.

5. As proposed in a previous round of revision, consider inserting a "Patient and public involvement" sub-section after the Data analysis sub-section, stating whether members of the public, people with lived disease experience of acute illness management in HaH. If you do not plan to include them, then simply write "None" under the "Patient and public involvement" sub-title. As highlighted in the previous round of revision, consider also inserting a "Protocol deviation"

sub-section just before the sub-section "Patient and public involvement" to specify how you would deal with any protocol modification after the publication of this manuscript. It is important to remind that protocol deviation is possible and this has been done since this protocol article has entered the review process. So, I find it appropriate to plan ahead.

Response:

Thank you for the suggestion. We have revised the manuscript as requested:

• A “Protocol deviation” sub-section has been added before the Patient and public involvement section, outlining how potential modifications to the protocol will be addressed in future publications (line 180-183, page 11)

• A “Patient and public involvement” sub-section has also been inserted, stating that no patients or members of the public were involved in the development of this protocol (line 185-186, page 11)

We appreciate your guidance in improving the manuscript.

6. Other comments

References.

I would delete reference 14 and use reference 23 for backing up the comment that was backed up by ref 14. Reference 24 is the same as reference 16.

Response: Thank you. Reference 14 has been removed and replaced with reference 23. Reference 24 has been deleted.

Line 149: consider replacing "using" by "by".

Response: Completed

Line 151: consider writing: "in a tabular form".

Response: Completed

Line 199: consider replacing "in the home" by "at home".

Response: Completed

Consider adding conflicts of interests and data availability statements before the reference section.

Response: Thank you for the suggestion. We have now included the “Conflicts of interests” and “Data availability statement” sections prior to the reference section, as recommended. See line 239-244, page 13

It is unclear to me whether you implemented Reviewer 3' s comment about the protocol registration on the OSF. I am unable to access the review protocol using the doi number provided in lines 38 and 92. Consider addressing this issue.

Response: Thank you for drawing our attention to this. We have completed the registration of the review protocol on the Open Science Framework (OSF), as suggested by Reviewer #3. As you have pointed out the DOI number was incorrect. We have now corrected the DOI number accordingly in line 38, page 2; line 92, page 5 of the manuscript and in the PRISMA Checklist. The link is now accessible as intended.

Reviewer #5:

Thank you for the opportunity to review the revised manuscript. I have limited my comments to my own concerns but note that other reviewers have generally responded positively to the most recent revisions which they requested. I have recommended "accept" but there are a few very minor adjustments that could be made pertaining to the search methods and strategy.

Response: Thank you for your positive assessment and for recommending acceptance of the revised manuscript. We appreciate your thoughtful review and have carefully considered your remaining suggestions regarding the search methods and strategy. We have made the minor adjustments as recommended to ensure clarity and completeness.

I disagree with the comment "We will search CENTRAL rather than the full Cochrane Library as our interest is limited to peer-reviewed original studies which CENTRAL specifically indexes". CENTRAL includes preprints and records from clinical trial registries, which would not be peer reviewed.

Thank you for this helpful observation. We agree that CENTRAL includes records from clinical trial registries and may also index preprints or studies that have not yet been peer-reviewed. Our intention in selecting CENTRAL was based on its comprehensive indexing of randomized and controlled trials, which aligns with our focus on identifying original research.

However, we acknowledge that CENTRAL does not exclusively contain peer-reviewed studies. To address this, we have clarified our rationale in the manuscript to reflect that CENTRAL is being searched to identify original research studies, and that we will apply additional eligibility criteria during screening to exclude non–peer-reviewed records such as preprints.

We have updated the manuscript to reflect this more accurate reasoning in line 106-114, page 6-7.

Within the search strategy, the following is noted:

exp Hospital to Home Transition/ (there are no subtrees so shouldn't explode - no effect on search)

By exploding "Scandinavian and Nordic Countries" - all but "Greenland" are already covered so listing them as separate MeSH is redundant. The only one required is "Greenland/"

Suggest adding truncation to pick up plurals for Norwegian*, Northern Europe*, and consider adding Swede, Swedes

Response: Thank you for your helpful suggestions. We have revised the search strategy accordingly – see Table 2 in the “Search strategy” section, page 7-8.

---

## [Editor Report · Decision Letter 5]

Mapping the organisational and interventional framework for patients admitted to Hospital-at-Home for acute illness in Scandinavia – a scoping review protocol

PONE-D-24-27648R5

Dear Dr. Hansen,

We’re pleased to inform you that your manuscript has been judged scientifically suitable for publication and will be formally accepted for publication once it meets all outstanding technical requirements.

Kind regards,

Mickael Essouma, M. D.

Academic Editor

PLOS ONE

Additional Editor Comments (optional):

Congratulations to the authors for your hard work that led to a substantial improvement of the quality of this protocol article.

Best wishes with your research

Mickael Essouma
---

## [Editor Report · Acceptance letter]

PONE-D-24-27648R5

PLOS ONE

Dear Dr. Hansen,

I'm pleased to inform you that your manuscript has been deemed suitable for publication in PLOS ONE. Congratulations! Your manuscript is now being handed over to our production team.

Kind regards,

on behalf of

Dr. Mickael Essouma

Academic Editor

PLOS ONE